# Retrieval of volcanic SO$_2$ from HIRS/2 using optimal estimation

Georgina M. Miles[1,2], Richard Siddans[2], Roy G. Grainger[1], Alfred J. Prata[1,3], Bradford Fisher[4], Nickolay Krotkov[5]

5  [1]Atmospheric, Oceanic and Planetary Physics, University of Oxford, Oxford, UK
[2]Remote Sensing Group, STFC Rutherford Appleton Laboratory, Harwell Oxford, UK
[3]Nicarnica AS, Lysaker, Norway
[4]Science Systems and Applications, Inc, 10210 Greenbelt Road, Suite 600, Lanham, Maryland, USA
[5]Goddard Space Flight Center, Greenbelt, Maryland, USA

10  *Correspondence to*: G. Miles (georgina.miles@stfc.ac.uk)

**Abstract.** We present an optimal estimation (OE) retrieval scheme for stratospheric sulphur dioxide from the High Resolution Infrared Radiation Sounder 2 (HIRS/2) instruments on the NOAA and MetOp platforms, an infrared radiometer that has been operational since 1979. This algorithm is an improvement upon a previous method based on channel brightness temperature differences, which demonstrated the potential for monitoring volcanic SO$_2$ using HIRS/2. The Prata method is fast but of limited accuracy. This algorithm uses an optimal estimation retrieval approach yielding increased accuracy for only moderate computational cost. This is principally achieved by fitting the column water vapour and accounting for its interference in the retrieval of SO$_2$. A cloud and aerosol model is used to evaluate the sensitivity of the scheme to the presence of ash and water/ice cloud. This identifies that cloud or ash above 6 km limits the accuracy of the water vapour fit, increasing the error in the SO$_2$ estimate. Cloud top height is also retrieved. The scheme is applied to a case study event, the 1991 eruption of Cerro Hudson in Chile. The total erupted mass of SO$_2$ is estimated to be 2300 kT $\pm$ 600 kT. This confirms it as one of the largest events since the 1991 eruption of Pinatubo, and of comparable scale to the Northern Hemisphere eruption of Kasatochi in 2008. This retrieval method yields a minimum mass per unit area detection limit of 3 DU, which is slightly less than that for the Total Ozone Mapping Spectrometer (TOMS), the only other instrument capable of monitoring SO$_2$ from 1979–1996. We show an initial comparison to TOMS for part of this eruption, with broadly consistent results. Operating in the infrared (IR), HIRS has the advantage of being able to measure both during the day and at night, and there have frequently been multiple HIRS instruments operated simultaneously for better than daily sampling. If applied to all data from the series of past and future HIRS instruments, this method presents the opportunity to produce a comprehensive and consistent volcanic SO$_2$ timeseries spanning over 40 years.

30  **1 Introduction**

Volcanic eruptions are important for climate and climate change. They perturb atmospheric chemistry and radiative transfer. Their signal in climatic records must be accurately quantified before any attribution of climate change to anthropogenic

sources. Furthermore, by studying the response of the atmosphere to volcanic eruptions in terms of climate sensitivity this can test ideas relating to climate prediction.

The monitoring of volcanic $SO_2$ emissions, the main precursor to sulphate aerosols, is crucial for accurately characterising total emission estimates but also for understanding plume evolution. Until the mid-1990's, only one principal instrument (the Total Ozone Mapping Spectrometer, TOMS) has been able to observe eruptions for an adequate period to generate something approaching a climate relevant record. The sensitivity of TOMS limits it to detecting only the larger, explosive eruptions rather than effusive ones where material remains predominantly in the troposphere. Satellite instruments that have been used to measure volcanic $SO_2$ are given in Table 1. From 1996, with the advent of the Global Ozone Monitoring Experiment (GOME) class instruments (UV-vis spectrometers) sufficient spectral resolution (and spatial resolution) has enabled the detection of lower amounts of $SO_2$ with higher accuracy from increasingly smaller eruptions. This has improved further still with instruments such as the Infrared Atmospheric Sounding Interferometer (IASI), from which $SO_2$, sulphate aerosol and ash may be derived simultaneously due to its high spectral resolution and broad spectral coverage (Karagulian et al., 2010). Total erupted mass estimates for volcanic eruptions can often differ by greater than 100% between instruments, as a result of sampling, geometry, differences in sensitivity and assumptions that contribute to algorithms, such as plume height. For example, Thomas et al., (2009) present a multi-sensor comparison of the 2005 eruption of Sierra Negra (Galapagos Islands), using concomitant observations by TOMS, OMI and MODIS. They found a wide estimate of total erupted $SO_2$ calculated from the three instruments, ranging from 60 kT to 1800 kT.

It is still the case that the operational period of these more sensitive, recent instruments is not yet long enough to constitute a climate-relevant record. Here we present the methodology for a relatively fast and accurate volcanic $SO_2$ detection and quantification method for an instrument originally designed to operationally measure water vapour and temperature profiles.

HIRS/2 has the potential to have captured stratospheric emissions from explosive eruptions continuously since 1979, but with significantly higher temporal sampling and greater sensitivity than TOMS. This enables the 35 year volcanic $SO_2$ emission record from satellites to be significantly enhanced, with potential uses for constraining models and examining in detail individual eruptions and plume evolution.

### 1.1 HIRS/2 Instrument

HIRS/2 is one of three instruments that originally constituted the Television Infrared Observation Satellite (TIrOS) Operational Vertical Sounder (TOVS), designed to provide atmospheric profile measurements of temperature and water vapour structure (Smith et al., 1979). The other TOVS instruments were the Stratospheric Sounding Unit (a radiometer) and the Microwave Sounding Unit (a scanning microwave spectrometer). The TOVS suite of instruments was first launched in 1979 aboard the new NOAA satellites based on the TIrOS-N design, and evolved in to the Advanced TOVS (ATOVS) system. Subsequent replacements have been deployed for the last 30 years aboard NOAA satellites (NOAA 6-17) (JPL,

2003), and more recently European platforms including most recently MetOp-A and B as HIRS/4. Throughout its deployment there have been at least two instruments (and occasionally three) orbiting simultaneously. HIRS/2 has 19 detector channels in the infrared and one in the visible part of the spectrum for cloud detection during the day. These channels are relatively broad, spanning between 0.1 and 0.5 μm depending upon wavelength. The key instrument parameters
are given in Table 2.

Two HIRS/2 channels coincide with $SO_2$ spectral absorption features, these being 7.3 μm (a strong asymmetric stretch vibration band) and 8.6 μm. The precise central wavenumber is dependent upon instrument version, and only HIRS aboard NOAAs 10 and 12 featured an 8.6 μm channel. These channels were originally chosen to be sensitive to water vapour for
use in sounding and applying corrections for the $CO_2$ and window channels. The 8.6 μm channel is also reported to be sensitive to volcanic ash and other aerosols (Kearney and Watson, 2009).

Channel 11 from HIRS/2 aboard NOAA11, centred on 7.2 μm, is shown in Fig. 1. Also shown are simulated transmission spectra for water vapour (which this channel was designed to detect) and $SO_2$, for two column amounts (1 and 300 DU). It
demonstrates both that the channel and spectral feature coincide well, and for large column amounts of $SO_2$ the channel would be strongly affected.

## 1.2 Previous efforts to retrieve of $SO_2$ with the HIRS instrument

Prata et al., (2003) demonstrated a method to detect volcanic $SO_2$ from HIRS, providing the $SO_2$ perturbation is strong enough, and located above any significant sources of water vapour. It is based on a synthesis of the expected clean
atmosphere brightness temperature for the channel, and the observed deviation from it when contaminated by $SO_2$. This method, hereafter referred to as either the Pratafit method or after Prata et al., (2003), uses a linear interpolation between the brightness temperatures of adjacent channels. It also assumes a fixed height of erupted volcanic $SO_2$, since theoretically only one piece of information can be obtained from one channel, and column amount is not insensitive to the height of the plume. The technique requires the $SO_2$ to be located in the upper troposphere/stratosphere above most of the atmospheric water
vapour, and there is no information about the height of the plume from the instrument itself. This information may be gleaned from other types of observations, but the fit is reliant upon the accuracy of this independent information.

A description of how the Prata method operates is detailed in Prata et al., (2003). While useful in itself, its most significant shortcoming is that due to its simplicity, the model is unable to capture atmospheric variability (other than potentially that of
$SO_2$). This particularly alludes to the variability of cloud, temperature and water vapour. Without independent height information of the $SO_2$ the radiance relationships are subject to potentially significant error. Indeed, it is not possible to formally quantify error of mass estimates from this method as it currently stands. Its strengths are that the operations required are computationally inexpensive and straightforward, as it is based on the principles of a band model. It has also

performed well against other observational data sets, although the previously mentioned uncertainties that contribute to error make quantifying overall uncertainty difficult. It uses a minimum offset threshold in brightness temperature for the channel affected by $SO_2$ in order to predict the presence of $SO_2$ and yet excludes the effects of atmospheric water vapour variability. As such, its sensitivity to low amounts of $SO_2$ is limited.

Guo et al. (2004) presented a re-evaluation of the 1991 Pinatubo eruption using $SO_2$ derived from HIRS/2 using the Prata-fit method, and compared it to SO2 derived from TOMS measurements. They were found to be broadly consistent. The Prata-fit method works sufficiently well to suggest that the 7.3 μm $SO_2$ feature it uses is robust enough to make further exploitation more refined. Use of information arising from other HIRS channels would constitute an improvement to the Prata fit

method, as multiple wavelength information can be used to diagnose attributes of the atmospheric profile such as temperature and the presence of cloud. This problem is well suited to an optimal estimation retrieval, which would incorporate a forward model (FM) of sufficient complexity to represent these atmospheric attributes. As with the Prata fit, unavoidably it will require some estimate of the altitude of an $SO_2$ plume.

**1.3 Outline of paper**

In Section 2, an Optimal Estimation (OE) retrieval algorithm methodology to extend the Prata-fit method is presented. Section 3 comprises an error study and presents results of retrievals from simulated measurements in order to understand the sensitivity of the algorithm and potential sources of error. Section 4 presents a case study of the 1991 Cerro Hudson eruption, where the algorithm is applied to real data and new eruption mass estimates are evaluated, and compared to

existing mass estimates from other instruments/methods. In Section 5 the results are discussed and further work is suggested.

**2 Methodology**

**2.1 Retrieval algorithm and forward model**

The HIRS/2 measurements used here are all-sky brightness temperatures from the instrument aboard NOAA11. This was

selected to demonstrate the capability of this version of the instrument with only one channel that is sensitive to $SO_2$ and two window channels that have some potential to be used to flag cloud and under some circumstances ash if required (although only one is used here directly). The brightness temperatures are a product derived from the raw voltage measurements via a radiance and brightness temperature conversion and have been subject to calibration factors and some basic quality control. Further information about the instrument is available from NOAA (1981) and elsewhere. The data format contains the time

in seconds from midnight of the measurement, the solar zenith angle, 19 IR channel brightness temperatures, one visible channel albedo, latitude, longitude, satellite altitude, line number for each orbit and the scan position (see Table 2).

Retrievals are obtained using the Levernburg-Marquart minimisation method after Rodgers (2000), and the full optimal estimation scheme used here is described in detail in Miles et al., (2015). The retrieval uses three HIRS/2 channels to derive three products: the $SO_2$ column, a scaling factor for a water vapour profile and effective cloud top pressure. The 7.3 μm channel is sensitive to both water vapour and $SO_2$. This channel may be said to saturate for $SO_2$ columns above 600 DU where significant increases in $SO_2$ result in small changes in channel BT below the envelope of the channel noise and other error terms. The weighting function for water vapour of the 6.8 μm channel peaks at around 500 hPa (around 5 km), and as such would have some sensitivity to the region where the vast majority of the water vapour in the column resides. To represent both channels accurately, some knowledge of cloud is required, which may be gleaned from the 11.1 μm channel window channel. This channel is highly sensitive to the emitting temperature of the lowest surface it observes (be it cloud or the surface), thus with some knowledge of the surface and atmospheric temperature profile it is possible to obtain an estimate of cloud top height. Other atmospheric gases not retrieved but contribute appreciably to channel brightness temperature are represented in the forward model by a climatological value. The potential error that this can introduce is incorporated into the estimate of forward model error.

Radiative Transfer for TOVS (RTTOV) is a radiative transfer model (developed by the UK Met Office, Saunders et al., 1999, ECMWF 2001) designed to simulate the instruments of TOVS including HIRS/2, and is used extensively (particularly for assimilation) because of its speed. It calculates layer transmittances for a variety of trace gas species using look-up tables of parameterised regression coefficients for a range of temperatures and pressures. It has been further developed since the TOVS system was first deployed, and version 10 is used here. RTTOV also has the functionality to compute partial derivatives.

RTTOV estimates channel brightness temperature based on pre-calculated coefficients for layer transmittances that are generated for a range of atmospheric profiles. As such, it is extremely fast, but as it stands it does not incorporate any representation of $SO_2$ other than at a very low climatological value. To alter the transmittance model to include $SO_2$ would require substantial re-working of program code. It is possible to calculate a set of predictor coefficients for $SO_2$ and incorporate them within RTTOV by replacing the properties of another gaseous species that has negligible impact on the total column transmittance within the selected HIRS/2 channels (in this case, carbon monoxide). The coefficients were generated by a `training' methodology using an extensive range of specimen atmospheric profiles, where the $SO_2$ was represented from very low/background levels to very large perturbations, after Matricardi (2008, 2010) and Siddans (2011). This approach retains the speed and accuracy offered by RTTOV and enables the model to be used to represent atmospheric gases for future instruments not already catered for (ECMWF, 2001).

For this work, the predictors were trained using profiles with up to 300 DU. Some care is required in the generation of these coefficients for $SO_2$. They are required to be limited to those that represent a first order relationship with $SO_2$ since the more complicated (higher order) predictors caused erroneous results. This is thought to be a result of both the dynamic range $SO_2$ can exhibit in a volcanically perturbed atmosphere, and the fact that RTTOV was not explicitly designed to model $SO_2$ for this instrument. The cost in terms of accuracy over this range of $SO_2$ is shown to be small, as demonstrated in Fig. 2 to be discussed in detail later.

The column retrieval developed here uses atmospheric profiles from the ECMWF ERA-Interim product (Dee et al., 2011) to represent atmospheric properties other than $SO_2$, or as a first guess in terms of the water vapour profile. These contain profiles on a pressure grid of 37 levels from 1000 hPa to 1 hPa. RTTOV is capable of generating weighting functions, but they refer to the sensitivity of the simulated measurements to perturbations in the atmospheric profile, rather than directly to changes in state vector. As a result, these are evaluated numerically in the forward model by successive FM calls where each element of the state vector is fractionally perturbed in turn. RTTOV has certain physical limits for its input values, and when occasionally the predicted updated state lies outside these they are constrained in the FM by the physical limits that RTTOV will accept, or that are appropriate for the forward model. These are 0.01 to 800 DU for $SO_2$, $1e^{-6}$ to 16 times the column water amount predicted by ECMWF and a maximum cloud top height of 16 km (a conservative upper limit for tropopause height). The weighting functions are allowed to make linear extrapolations beyond these limits, allowing the retrieval more freedom, but unphysical profiles are suppressed with quality control of the derived products (discussed later).

## 2.2 Profile definition in forward model

In the absence of any further information, an effective $SO_2$ profile must be represented in the forward model. The three-element state vector comprises a scaling factor for the $SO_2$ profile, a scaling factor for a water vapour profile and a cloud top pressure. A volcanic $SO_2$ perturbation is represented by a vertically localised triangular profile. This triangular profile is normalised to have an integrated mass of 1 DU. This was partly done to ease interpretation, since the retrieved scaling factor would be approximately equal to the total amount of $SO_2$ in the column. The rest of the profile is prescribed by a background $SO_2$ volume mixing ratio climatology, the total column mass of which is less than 1 DU. In the forward model, a scaling factor applies to a specified height region of the $SO_2$ profile, scaling all elements within and none outside this. The expected region of the volcanic plume is estimated using ancillary information, such as lidar or results from modelling of the eruption available in the literature. The sensitivity to how well the altitude and thickness of an $SO_2$ plume is evaluated using retrievals from simulated measurements, and detailed in Section 3.

In an analogous way to $SO_2$, $H_2O$ is represented in the state vector by a profile scaling factor, but it applies to the entire profile rather than a localised height region. The profiles used for retrieval are those collocated from the ECMWF ERA-

Interim product for a given HIRS/2 pixel (which represents the best guess for the state), but in principle any climatological profile can be used. In the case where a scaling factor is close to one, it would indicate that the $H_2O$ profile is similar to that which produced the measurement.

The third element of the state vector is cloud top height (CTH), or specifically the geopotential height at an equivalent pressure level. It was found that the speed of convergence was significantly reduced if the initial guess of cloud top pressure was reasonably accurate. As such, this is derived before the retrieval using interpolation between calls to a radiative transfer model that simulates the 11.1 μm channel brightness temperature (BT) for 0-10 km (using associated ECMWF ERA Interim temperature profile), and included a test for temperature inversions.

### 2.3 Error

An estimate of forward model error was calculated using the Reference Forward Model (RFM) — a line-by-line radiative transfer model (Dudhia, 2002), discussed further in Section 3. The estimate accounts for inaccuracies that arise due to modelling the atmosphere at reduced spectral resolution, limited vertical resolution (100 m versus 1 km as used in the forward model outside the region of the $SO_2$ perturbation), inclusion of non-retrieved trace gases at a climatological level or their preclusion entirely, relative to a reference case. This yields a channel quantity (in brightness temperature) that is combined in quadrature with the noise equivalent differential radiance for each channel, and is thus incorporated into measurement noise for the purposes of the retrieval. The a priori error associated with cloud height is 10 km. The a priori error for water vapour is based on the variance of water vapour in the ECMWF atmospheric training profiles discussed above relative to the mean.

### 2.4 Estimation of $SO_2$ and $H_2O$ covariance for HIRS/2

Establishing an appropriate $SO_2$ a priori error is potentially a non-trivial issue with regard to a retrieval problem where the measurements have relatively little sensitivity. A volcanically perturbed $SO_2$ profile can contain 2 or 3 orders in magnitude more than a background profile, and at the centre of a large plume this can be even more. A good a priori error gives the retrieval the freedom to find a correct minimum in cost space, and can restrict it from converging on a solution that is unphysical. The variance for a background profile would be very small, as opposed to a profile where $SO_2$ is expected, which would be very large. If there is sufficient information contained within the measurements, one would conventionally use a variance that spans both scenarios. This results in a poor constraint for an ill posed problem but is necessarily used here, where a first guess/a priori error of 100 DU is used and a prior variance is the first guess squared. 100 DU represents an $SO_2$ column from a large, explosive volcanic eruption. Pinatubo, for example, yielded column amounts of 350-500 DU (depending upon instrument) after 24 hours which reduced to 100 DU after 7 days (Carn et al., 2005). The OMI instrument (see Table 1) captured column amounts of around 200 DU after the 2008 eruption of Kasatochi (Prata et al., 2010).

Early results of the retrieval scheme run with real measurements revealed that there were many `false positives' of $SO_2$ retrieved. Their structure indicated that they were related to the presence of water vapour, or errors in the fit for water vapour. This indicated the degree of covariance between $SO_2$ and water vapour which had to be incorporated into the retrieval since the 7.3 μm channel is sensitive to both water vapour and $SO_2$.

The retrieval was applied to one day of `clean' measurements in the Southern Hemisphere where no volcanically perturbed profiles were expected. The retrieval was forced not to retrieve $SO_2$ by artificially constraining the a priori variance, but none-the-less small amounts of $SO_2$ are retrieved from that channel because of inadequacies in characterising the water vapour. The brightness temperature fit residuals in the $SO_2$ channel were very small, but it is expected that nearly all of the

$SO_2$ being retrieved on this day is being falsely attributed. The standard deviation of the 7.3 μm channel brightness temperatures fit residual in the retrieval of 0.92 K constitutes an estimate of the 'real world' error covariance of water vapour with $SO_2$ for this instrument. This is incorporated by adding it in quadrature to the forward model error for this channel and resulted in a significant reduction in the occurrence of false positives.

**3 Error study: Retrievals from simulated measurements**

There are some sources of error that can be incorporated and dealt with by the retrieval. These include measurement noise, the presence of cloud or ash, $SO_2$/$H_2O$ covariance and an estimate of forward model error discussed above. The main sources of error that cannot be adequately represented in the forward model are errors that impact ill-posed nadir $SO_2$ column retrievals in general. These are incorrect height assignment of the $SO_2$ plume, incorrect thickness in the plume represented in

the forward model and, particularly in the case of infrared measurements, sensitivity to the presence of cloud and/or water vapour. Their relative impacts vary and the sensitivity of the solution to them can be quantified using simulations. It should be noted that some of these errors (plume height and profile shape) cannot often be known at the time of retrieval, and as such the actual impact on the retrieval result also cannot be known. They are investigated here in order to give a general indication as to the potential error that can be associated with the results, to give a window of confidence. Others, such as

the impact of cloud or ash on the retrieved $SO_2$ error can be investigated for use in quality control.

**3.1 Spectral precision of forward model**

In order to assess the accuracy of the RTTOV-based fast column retrieval forward model, it is compared to simulations from a model with a higher accuracy. The RFM is a line-by-line radiative transfer model (Dudhia, 2002) capable of modelling the

atmosphere at a spectral resolution of up to 0.0001 cm$^{-1}$. The RFM is not suitable for the forward model because it is

computationally expensive and it does not inherently represent any effects of cloud or ash. Figure 2 shows the results of column retrievals from HIRS/2 channel BTs simulated by the RFM, using a sample ERA-Interim cloud-free meteorology (temperature and water vapour profiles) at 0 and 60°S latitude and 0°W longitude, where only the column amount of $SO_2$ is changed in the simulation. It also shows the $SO_2$ fit by the Pratafit method. The Pratafit method does not fit $SO_2$ below 5 DU, which depending upon the atmospheric state can be equivalent to an observed brightness temperature difference of up to 4 K. The bias of the Prata fit has a dependence upon latitude, primarily because of the different amount of water vapour in the profile at the two latitudes shown here. The column retrieval has a very small bias that only becomes perceptible at $SO_2$ loadings approaching 200 DU, at which point it is of the order of <5 DU.

## 3.2 Sensitivity to forward model representation of $SO_2$ plume

Both the altitude and amount of $SO_2$ affect the 7.3 μm channel brightness temperature but as there is only one channel sensitive to $SO_2$ on NOAA11 considered here, there is at most one piece of information that can be retrieved for $SO_2$. Therefore, for an accurate retrieval of $SO_2$ column, it is important to have some knowledge of the plume altitude or its vertical profile. The column retrieval developed here requires some information of the height of the $SO_2$, but this can be subject to uncertainty and may change with time. As such, the sensitivity of the retrieval to errors associated with plume height and specification must be examined.

### 3.2.1 Altitude

Measurements were simulated for a plume at a range of altitudes from 8-18 km. Figure 3 shows the impact on the retrieved $SO_2$ column at a specified, fixed altitude of 12 km as a fraction of the true column at these altitudes. Errors range from typically ±0-30 % for most column amounts up to 100 DU, and increase for larger amounts, and for particular altitudes. While the specific error may be state dependent (upon meteorological conditions, specifically the water vapour profile), these simulations do give a general indication as to the magnitude of error that can result from incorrect height assignment of the volcanic plume in the forward model. This is the largest source of error in the OE column retrieval (and the Prata-fit method) and is made more challenging because there is a dependency of the error on column amount. Since height assignment errors cannot be known such simulations can at least give a general indication of potential uncertainty of retrieved amounts, depending on the quality of information available regarding altitude of volcanic $SO_2$. It is clear therefore that good prior knowledge of the $SO_2$ plume altitude is necessary for accurate retrieval or fit of $SO_2$ column amounts from HIRS/2.

The performance of the column fit was also directly assessed against a line-by-line model (RFM) for plume altitudes from 8 to 18 km (where the plume height assignment used in the retrieval was the same as that used in the measurement simulated by the RFM) and it was found that for altitudes of over 17 km the column fit was unable to retrieve $SO_2$ columns less than 30 DU, but in all other cases true clear-sky column amounts were retrieved accurately from simulated measurements.

### 3.2.2 Profile shape and plume thickness

Figure 4 shows the consequences that can result from retrieving the volcanic plume with a fixed profile shape that represents the thickness of the plume incorrectly. Measurements were simulated using a triangular profile centred at 12 km but with

10 baselines of 1 and 4 km. They were then used in the retrieval with a fixed profile shape with a triangular perturbation also centred at 12 km, but with a baseline of 2 km (thought to be the best representation of the plume used in the case study in Section 4). The retrieval simulations suggest that errors are larger when the plume thickness is overestimated (typically 13 %), with only small inaccuracies introduced when the plume thickness is under-estimated (less than 2 %). The modelled cloud top height was 3 km in all cases. It is therefore possible that an underestimate of plume thickness would result in

smaller errors.

### 3.3 Sensitivity of Retrieval Scheme to Cloud and Ash

Some understanding must be obtained of how the column retrieval forward model behaves in the presence of ash and cloud of different type. The forward model fits a cloud top pressure using the 11.1 μm channel, which is expected to work well for

most scenes with cloud in the troposphere. The effect of cloud on the other channels is examined here using a cloud model, the Oxford-RAL Retrieval of Aerosol and Cloud (ORAC) model. The model is described in detail by Poulsen et al. (2012), where it was used as part of an optimal estimation retrieval of cloud properties for the Along Track Scanning Radiometer (ATSR) by simulating radiances in a combination of visible, near infrared (NIR) and IR channels. The model parameterises a cloudy scene by ascribing cloud phase, effective radius of a size distribution, the 0.55 μm optical depth and a cloud top

pressure. It uses the plane parallel approximation and models cloud as a single layer. The model represents trace gases at a background climatological level. The system can also be used to retrieve ash plume properties: plume height, optical thickness and ash particle effective radius (McGarragh et al., in preparation, 2017)

HIRS/2 measurements were simulated for a range of liquid and ice cloud and ash optical depths, effective radii and at a range of altitudes when no volcanic $SO_2$ is present. These channel brightness temperatures were then used to retrieve $SO_2$ to

30 identify where this resulted in an erroneous fit.

An example is shown in Fig. 5, which shows that for liquid water clouds above 5 km, the column retrieval erroneously retrieves some $SO_2$ when there is none, the water vapour and cloud top height become inaccurate and the fit cost begins to

increase. The results indicated that low optical depth or effective radii for cloud or aerosol can result in poor fitting of the measurements, both resulting in an underestimate of cloud top pressure with false positives of $SO_2$ and an over-estimation of water vapour. This yields a crucial quality control threshold where retrieved cloud top altitudes of greater than 5-6 km should not be trusted, as they are likely to result in spurious detection of $SO_2$ and a high retrieval cost. This may imply that very thin

cloud beneath 5 km (or incorrectly retrieved to be) could still contribute to poor fitting of the measurements.

## 3.4 Quality Control

The results of the column retrieval must be subject to some quality control.  In addition to the disregard of non-converged

and converged pixels with cloud retrieved at an altitude greater than 5 km, a retrieved column is only considered useful if the error is less than the retrieved amount.  Quality control becomes very important when erupted plumes are used to calculate total erupted mass, where even a small amount of noise can yield a biased mass total.  For the purposes of gridding or summing pixels for deriving a global/plume mass estimate, a minimum retrieved $SO_2$ threshold may be applied in deference to the lower detection limit of the retrieval, in order to avoid spurious low values that the retrieval should not be sensitive to,

such as those relating to water vapour or cloud that are not accounted for in either the error covariance or the forward model. An effective way of obtaining this quantitatively is to apply a 2 or 3 sigma test, where sigma is the standard deviation of the retrieved $SO_2$ on a day when no volcanic $SO_2$ is expected to be present. This threshold gives statistical confidence that a value above it is significantly distinct from the noise above the 95 or 97 percentile. The sigma threshold for 6th August 1991 (a day when there was no $SO_2$ present in the region relating to the case study in Section 4) was 2.7 DU, and is probably a

lower estimate of the detection limit of the HIRS/2 $SO_2$ column retrieval in the mid-latitudes. Multiples of this value indicates confidence that a retrieval result is dominated by signal rather than noise.

## 4 Case Study: Cerro Hudson Eruption in 1991

Cerro Hudson (45.54°S, 72.58°W, elevation 1905 m) is a stratovolcano in the south Chilean Andes that erupted explosively in August 1991, two months after the Pinatubo eruption.  The eruption was estimated to be 10-20 times smaller than Pinatubo in terms of $SO_2$ that was expected to be emitted.  In this sense, as well as being a non-equatorial eruption, it has similarities to the 2008 Kasatochi eruption in the Northern hemisphere.  It is selected here as a case study because it was a relatively large eruption that has not been studied exhaustively, and a very good example of an eruption in recent satellite

history which only TOMS observed with any significance, that can benefit from application of this technique.

At the time of the 1991 eruption, the only satellite available that could detect $SO_2$ with any demonstrated accuracy was TOMS. The Microwave Limb Sounder, a contemporaneous instrument that observed $SO_2$ from Pinatubo at a higher altitude, produced noisy results in the lower stratosphere at this latitude (Read et al, 1993). In addition, contemporary lidar measurements of the Hudson plume were made at the CSIRO (Commonwealth Scientificand Industrial Research

Organisation) Division of Atmospheric Research, at Melbourne, Australia (38 S, 145 E) (Young et al., 1992, Barton et al. 1992). These measurements are sensitive to ash, sulphate aerosol and meteorological (water) cloud. The backscatter profiles tend to indicate peaks at around and above 20 km, and frequently at 10-13 km. The higher peak is attributed to aerosol from the Pinatubo eruption. Young et al. (1992) interpret the majority of observations that are thought to include Hudson material as the feature at 12 km in October, with variable cirrus at 10 km. It is reported by the authors that the plume

was observed consistently from 28th August until December 1991 between 10 and 13 km, with a decreasing scattering ratio. The relative proportions that contribute to the backscatter measured are expected to be dominated by ash in the first few weeks after the eruption. Little ash is expected to be present after a month beyond the eruption, but by this time the vast majority of the $SO_2$ will have oxidised into aerosol. Whilst lidar is not sensitive to the presence of gaseous $SO_2$, inferences can be drawn from the height of the aerosol it eventually becomes. In this case the lidar information is considered to be a

valuable starting point as a guide for estimating the cloud height of the $SO_2$, in the context of other information. As well as some ground observations, the Hudson eruption was sensed remotely by AVHRR (ash), lidar (sulphate aerosol) and incidentally by an aircraft (Barton et al. 1992). Hofmann et al. (1992) reported possible exacerbation of Antarctic ozone depletion of 10-20% of total column due to the presence of Hudson aerosol in the lower stratosphere for September 1991. The anomalous depletion occurred within the polar vortex predominantly at 11-13 km and 25-30 km, the respective altitudes

of the Hudson and Pinatubo aerosols.

The transport of the Hudson volcanic plume was first numerically modelled by Barton et al. (1992), to reasonably good agreement with satellite and lidar observations. The plume was also modelled using an isentropic trajectory model, initiated by TOMS observations of $SO_2$ (Schoeberl et al. 1993). These models showed good spatial agreement with observations for

the first eight days after the eruption which is an indication that the height assignment of the erupted plume was accurate within the models. The most explosive eruption began and ended on 15th August. It was at this stage of its eruptive phase that the majority of the material was injected into the stratosphere (Constantine et al., 2000).

### 4.1 Results

Using all of this information, the Hudson plume is modelled as a triangular peaked profile with a baseline of 2 km between

11 and 13 km, peaking at 12 km. Figure 6 shows an example of the $SO_2$ retrieval applied to a day of data on 15[th] August 1991, and its associated retrieval error. Figure 7 shows results for the same day as Figure 6, but for the other elements of the state vector: the retrieved water vapour scaling factor and cloud top height (with their associated retrieved errors). Only high cost and convergence criteria have been applied. In general, the retrieved values of cloud top height have very small errors.

For the water vapour scaling factor, the largest errors occur in the presence of high or thick cloud, which is expected. As shown in Section 3, the cloud model simulations suggested that the retrieval struggles in the presence of high cloud and can on occasion fit spuriously enhanced $SO_2$, potentially because it results in a poor estimate of water vapour in the correspondingly colder scene. Regions of very high water vapour scaling factor result in very high errors in retrieved $SO_2$,

and data with cloud top height greater than 5 km are not considered reliable for $SO_2$.

Figure 8 shows nine days of retrieved $SO_2$ from the 1991 Cerro Hudson eruption following the largest eruption phase on 15[th] August. The eruption began on 8[th] August emitting smaller amounts of $SO_2$ into the upper troposphere lower stratosphere, which can be seen as already present in the path of the main plume on subsequent days. The multiple sampling of the plume

by successive orbits (day and night) is quite apparent, particularly as the plume becomes more distorted after 20[th] August.

## 4.2 Plume Mass Estimate

The simplest method to estimate the total erupted mass or mass present in a volcanic plume is to take the sum of the representative footprint areas of the satellite that measured $SO_2$. This method presents several problems relating to sampling of a volcanic plume, particularly with an infrared instrument that measures both night and day that could sample the plume

more than once, orbits may partially sample the plume in any one swath and the plume will move constantly between sampling. Alternatively, gridding averages the data into grid boxes on a latitude and longitude grid. Some care must be taken to account for whether or not the gridded data are representative of the data resolution, and keeping track of bins with no data can be a way to estimate under-sampling. Guo et al. (2004) used two methods of gridding data, that of kriging for TOMS data and nearest neighbour interpolation for HIRS/2 (Pratafit method) to account for larger spatial gaps between

points. These methods either impose statistical methods or manually introduce information based on assumptions. While both can be utilised in such a way as to indicate an estimate of the error or uncertainty that this introduces, mass estimates presented here are only based on the sum of equivalent contiguous footprints represented by each HIRS ellipse.

Furthermore, if gridding is used, in order to ensure that the data are sampled fairly, the orbits should first be split into

ascending and descending nodes, with care taken regarding where a plume is in relation to the date line. This is in an effort to minimise recording the same data point twice when the plume has moved by the time the region is sampled again. Other methods are available but often require a model or further ancillary information.

## 4.3 Comparative measurements of $SO_2$

The plume mass estimate for the HIRS/2 $SO_2$ retrievals for the Cerro Hudson eruption may be qualitatively compared to the figures for TOMS within Constantine et al. (2000). Total erupted mass estimates given can be directly compared, as shown

in Table 3, although the methodology by which the estimates were derived differs. Spatially, HIRS/2 has the advantage of smaller footprint than that of TOMS, (IFOV 1.25° x 1.25°/17.4 km x 17.4 km versus 3° x 3°/ 50 x 50 km) but the TOMS swath is 50% wider (3000 km). For a case such as the Hudson plume, TOMS is more likely to capture the entire plume in one orbit swath and sample it only once, which on the one hand greatly reduces ambiguity in deriving total plume mass but on the other hand the frequency of observation is reduced and sometimes only part of the plume is captured. As reported by Constantine et al. (2000), this was sometimes the case, and a 'best' estimate of the TOMS data was used to contribute to the values in Table 3.

The erupted mass estimates given in Table 3 that relate to HIRS/2 are the sum of equivalent footprint areas , from nodes that capture the most of the $SO_2$ plume present each day. Figures are rounded to reflect probable accuracy. For the total eruptive period, this method has yielded a total erupted $SO_2$ mass estimate of 2300 kT with an averaged retrieved error of 27 %. This error does not incorporate error that arises from uncertainty in the height of the $SO_2$ in the forward modelled plume (as demonstrated in Section 3), or error that might arise from discounting pixels where $SO_2$ was retrieved below the 3-sigma threshold. It does not account for absent scanlines due to instrument calibration, so should be considered a lower limit. As previously discussed, a good estimate of plume height is an unavoidable requirement in $SO_2$ detection with an instrument with only one channel sensitive to atmospheric $SO_2$. In the case of this work, height assignment error of $\pm$ 1 km introduces a mass dependent bias of between 5 and 20% for a given pixel depending upon where in the atmosphere the plume is located. For TOMS, the approximate error suggested for the total erupted mass estimate is 30% (Krueger et al, 1995, Constantine et al. 2000).

The TOMS algorithms used in Constantine et al. (2000) have been recently updated, and a brief comparison is presented here to some initial data from an updated TOMS algorithm. This algorithm exploits the way ozone and sulphur dioxide both strongly absorb UV radiation. The new TOMS algorithm builds on the early heritage of BUV algorithms (Krueger et al., JGR, 1995). These algorithms retrieve both $O_3$ and $SO_2$ by taking advantage of the large $SO_2/O_3$ cross section ratio (CRS) differences in the gas absorbing bands. This approach constructs radiance tables using a forward model that accounts for both the $O_3$ and $SO_2$ cross sections. The new algorithm uses the 317 nm channel to retrieve $SO_2$ (CRS ~ 2.5), the 331 nm channel to retrieve O3 (CRS ~0.15), and the channel at 340 nm to retrieve the spectral dependence, dR/dλ. This methodology further applies a small second order step2 correction that accounts for non-orthogonality between the $SO_2$ and $O_3$ channels.

A one week composite of retrieved $SO_2$ for both instruments is shown in Fig. 9 where $SO_2$ from the main eruptive phase can be seen circumnavigating the hemisphere. There is clear complementarity between the instruments in terms of absolute amount retrieved and characterisation of the plume. The smaller pixel size of HIRS and more frequent sampling enables the plume to be observed in finer detail; however the wider swath of TOMS frequently captures more of the plume in one swath.

For a more detailed comparison, two orbits during the 1991 Hudson eruption are considered where the plume is almost fully sampled by both instruments, as shown in Fig. 10. The pixels in the region of the plume were also relatively cloud-free or had low cloud during the observation.

The geographical bounds considered for the mass estimate are between -53° and -45° in latitude and 10° to 60° in longitude. Using the method of summing over mass and area discussed previously, the mass of the plume represented here by HIRS/2 and TOMS is calculated to be 1398 and 1540 kT respectively, after quality control has been applied. The missing four scan lines due to a HIRS calibration phase that coincide with the plume in the region of high concentration suggests the HIRS estimate is an underestimate. It is apparent that HIRS/2 is potentially more sensitive to lower amounts of $SO_2$. It is
challenging to directly compare the $SO_2$ retrieved by two instruments with differing footprint sizes. Gridding might offer an alternative method of plume mass estimate, but selection of the most appropriate grid box size relative to the pixels of each instrument coupled with the small size of the plume with a strong $SO_2$ concentration gradient make it a challenge for such a comparison to be equitable and account for instrument attributes. A comparison involving gridding for a larger eruption (c.f. Pinatubo) would be less problematic.

### 4.4 E-folding time

The e-folding time for erupted $SO_2$ is a measure of the residency of the material in the atmosphere, and is affected by the height the material reaches and in the case of very large eruptions, the amount itself. It is also affected by wind shear (horizontal and vertical) and humidity, which affects the rate at which the $SO_2$ is oxidised and sulphate aerosols grow. The
20 measure is more suited to large eruptions (e.g. El Chichón in 1982 or Pinatubo in 1991), in terms of inferring effects upon radiative forcing, about which Miles et al. (2004) and other works are concerned. This is because the amount and height that such eruptions reach in the stratosphere gives the $SO_2$ sufficient time to become globally mixed, and as such affect the radiative forcing globally. Equation 1 describes the process of exponential decay, where $N(t)$ is a quantity at time $t$, $N_0$ is the initial quantity at time $t=0$ and $\lambda$ is the decay constant.

$$N(t) = N_0 e^{-\lambda t} \tag{1}$$

The e-folding time, the time in which the initial quantity is reduced to 1/e of its initial value, is given by the reciprocal of the decay constant. Using approximate values from the mass estimates derived from Fig.9 where the total $SO_2$ can be said to drop from around 1500 kT (the total mass present on 17[th] August 1991 associated with main plume) to 500 kT 18 days later, this yields an e-folding time of around 16 days. Two days after the largest plume was erupted is used here to minimise
potential obscuration of the plume by the coincident presence of thick ash. In reality the total mass observed does not decay smoothly, but has noise due to the fact that the plume is not always perfectly sampled, and the number of retrieved pixels excluded due to the presence of high or thick cloud or ash varies. The variability of the mass estimates and the associated

retrieval error make only an estimate appropriate for this approach, but it is not considered to be an unreasonable one. If the e-folding time is calculated for the extremes of the retrieved error bounds of the mass estimates, the e-folding time is 10 days at a minimum, and 35 days at its shallowest descent, but these are considered to be overly-generous bounds by this method. This case is complicated by the fact that about 30 % of the $SO_2$ released by Hudson was erupted over the 7 days before the main eruption on 15th August, making the calculation of the decay subject to further uncertainty. The e-folding time for this $SO_2$ plume as estimated by Constantine et al. (2000) is around 15 days, but they state that this is algorithm dependent. These estimates are somewhat smaller than the e-folding times for the larger eruptions (e.g. Pinatubo), which is to be expected due to the considerably lower altitude of the Hudson plume. More recently, Carn et al. (2016) estimated the e-folding time of Cerro Hudson to be ~7 days, based on mass estimates from TOMS (Constantine et al., 2000). They attribute this anomalously short e-folding time to the late southern hemisphere winter timing of the eruption. However, since Constantine et al., (2000) estimate nearly twice the initial total mass (4000kT) than that observed by HIRS/2 in this work (and the subsequent TOMS algorithm discussed here) it is possible that the inconsistency in e-folding times could be due to an over-estimate of initial erupted mass from the original TOMS algorithms. Total mass estimates (and therefore e-folding time estimate) would be improved greatly in accuracy if the HIRS/2 instruments aboard NOAA10 and NOAA12 that were also present were used to result in very comprehensive sampling of this eruption.

## 5 Discussion

This OE column retrieval finds a new total erupted mass estimate for the 1991 eruption of Cerro Hudson of 2300 ± 600 kT from the HIRS/2 instrument aboard NOAA11, where the error is the retrieval noise. This does not incorporate any error from plume altitude estimation but the potential impact has been quantified by forward model simulations. This total mass estimate is lower than that of TOMS (Constantine et al. 2000) and that of Carn et al. (2016) but higher than that derived in a similar way using the methodology of Prata et al. (2003) for HIRS/2. Reasons for this include (but are not limited to) differences in sampling, height sensitivity, instrument differences and attributes or accuracies of the forward model or fit employed in $SO_2$ detection. From the comparison with the new TOMS algorithm, the HIRS/2 results presented here are highly consistent, and further quantitative comparison, for this eruption in particular, is desirable.

The retrieval precision demonstrated in this case study is slightly smaller (~3 DU) than that proposed for the TOMS instrument (6-7 DU). As such, with the increased sampling of the IR instrument it is apparent that HIRS/2 can offer a positive contribution to the atmospheric $SO_2$ emission record from explosive volcanic eruptions up to and beyond the launch of GOME and other satellites that followed. Moreover, benefits of the optimal estimation approach over and above the more rapid but limited brightness temperature difference method are significant. They include a quantified error on individual pixel retrieved values, latitudinal variation in accuracy, diagnostic indicators of the retrieval performance and goodness-of-fit and treatment of cloud and water vapour consistent to the retrieval of $SO_2$. When summing mass over a large number of

pixels, the precision that these afford becomes increasingly important. Issues that remain are those endemic to ill posed problems where there is only one piece of information on $SO_2$ available and only limited information about the height or shape of the profile of a volcanic plume. It is conceivable that further progress might be made by using HIRS/2 aboard NOAA10 and 12 with the addition of the 8.6 µm channel in ash-free pixels.

There are clear opportunities for extending this work. In particular, as the HIRS/2 instrument was present aboard a number of the NOAA platform series, and often simultaneously flown (NOAA 10, 11 and 12 were all in orbit at the time of the Cerro Hudson eruption) there is the possibility to fully characterise eruptions with very high temporal sampling. More rigorous methods for interpolation, sampling and gridding the data can also be used to reduce errors in the total mass estimates. The

application of further tools such as chemistry transport or trajectory models for understanding plume evolution would be better constrained by the availability of more measurements.

The first HIRS instrument was flown aboard TIrOS-N in 1978, and there are almost continual data available to the present, and for the foreseeable future of the Met-Op series of satellites, enabling a potential dataset spanning 40+ years. Generating an $SO_2$ dataset for the duration would be an opportunity to maximise the value and legacy of the satellite data. Such a data-

set, with an accompanying error covariance estimate could be used as input to a climate model to better assess the effects of large volcanic eruptions on the radiative balance of the atmosphere. For much of the latter half of that period, there are (and will be) other satellite instruments capable of measuring $SO_2$ in the limb and the nadir, in particular high resolution spectrometers with very much enhanced accuracy and precision, that will provide correlative information about the quality of the HIRS/2 $SO_2$ column retrievals that may be considered in retrospective terms. There is also a break in the TOMS

record during 1995–1996 that can be filled by HIRS/2 estimates.

It would be highly desirable to extend comparisons from this eruption with TOMS $SO_2$ in general, comparing a longer record by both instruments for other eruptions, since both provide a unique record of $SO_2$ potentially spanning many decades. Satellite records of this length for climatologically important trace gases are rare, and would also provide further constraint to volcanic $SO_2$ emissions in coupled chemistry climate models.

**Acknowledgements**

This work was in part funded by the Natural Environmental Research Council. The authors would like to thank Prof. F.W.

Taylor for his comments and suggestions regarding this work, and David Latham. Additionally the authors would like to thank STFC Rutherford Appleton Laboratory for its facilitation, and the Associate Editor Thomas von Clarmann for his helpful comments.

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

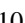

Figure 1: Transmission spectra of $H_2O$ and $SO_2$ simulated from southern hemisphere midlatitude water ECMWF ERA Interim background vapour profile using the RFM (see text). The $SO_2$ spectra were simulated using triangular profiles to represent column amounts of 1 and 300 DU, as used in the forward model.

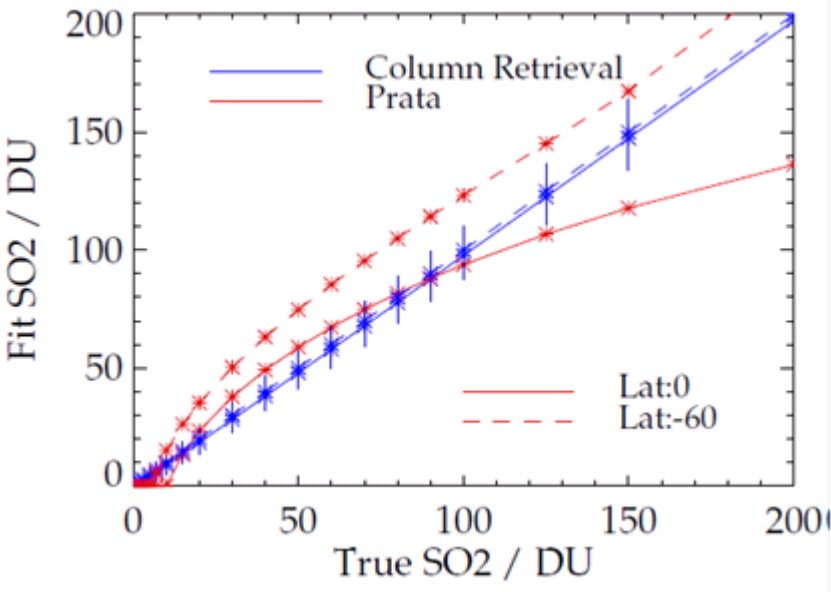

Figure 2: Retrievals based on simulations by a line-by-line model (RFM), with synthetic measurement noise. The error bars for the column retrieval are the retrieved errors. These simulations use temperature and water vapour from a cloud-free ECMWF ERA-Interim atmosphere on 15th August 1991, for a grid box centred at 0°E and both 0°N and -60°N and 0°E. The vertical bars show the retrieved error for the column retrieval. No error estimates are possible for the Prata fit method.

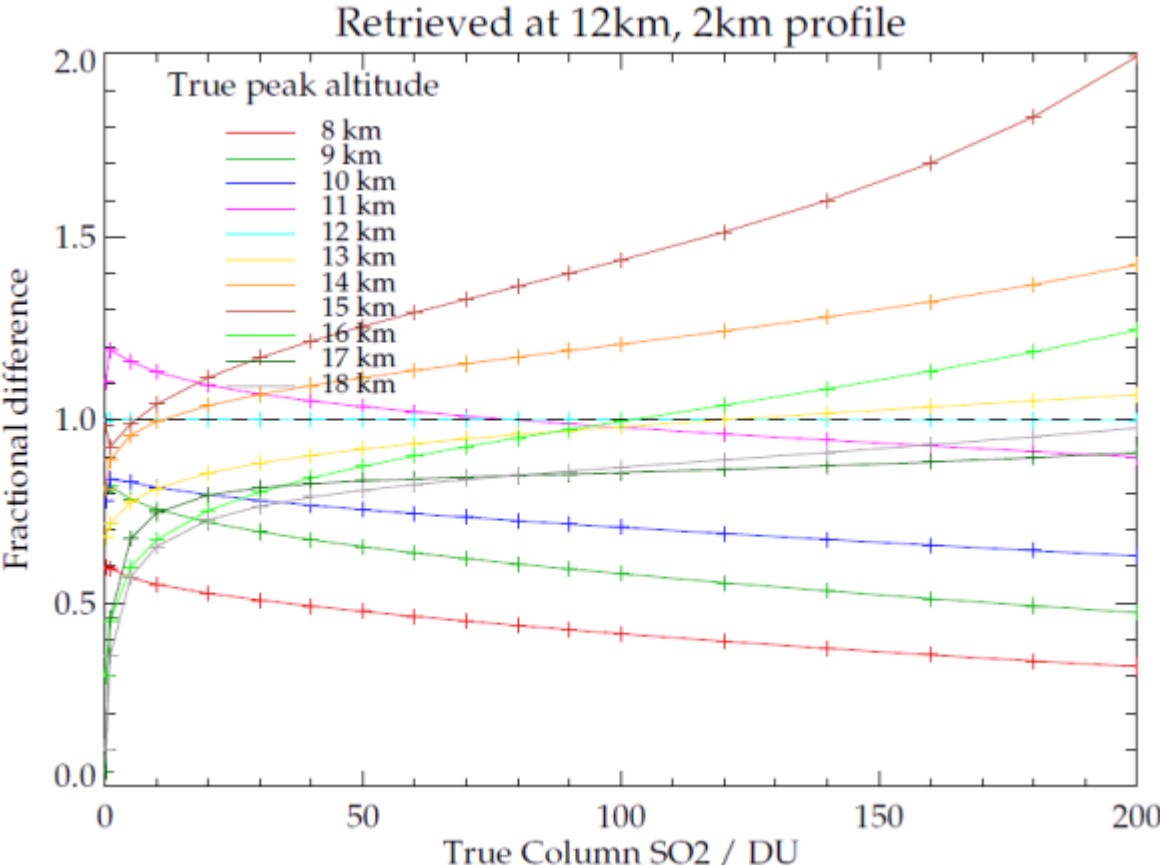

Figure 3: A measurement was simulated for a volcanic plume of triangular profile centred at a range of altitudes, for a range of total column amounts. A retrieval is then performed where the plume is assumed to be at 12 km. The fractional difference, or error, is plotted.

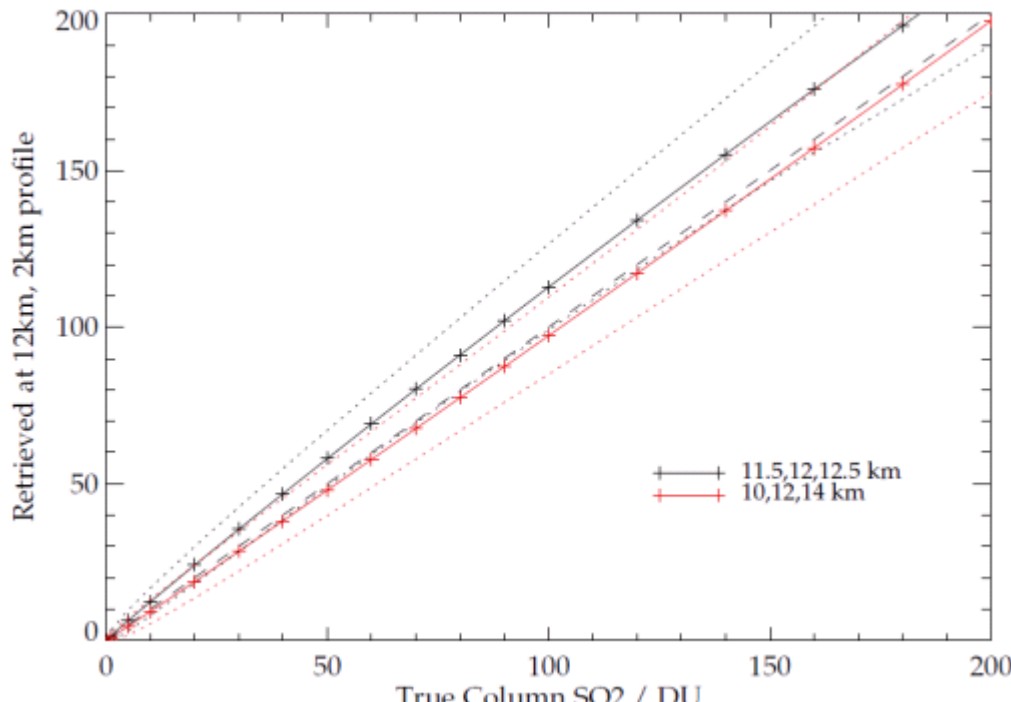

**Figure 4: The black line indicates how columns from 0.1-200 DU are retrieved on a fixed grid with a scalable triangular profile with base, mid-point and top at 11, 12, 13 km respectively, when the true profile shape is given by a triangular profile at 11.5, 12, 12.5 km, effectively over-estimating the thickness of the plume. The red line shows the equivalent result for an underestimate of the plume thickness, the real profile given by 10, 12, 14 km. The dotted lines show the bounds of retrieved error in each case. The dashed line is x=y shown for clarity.**

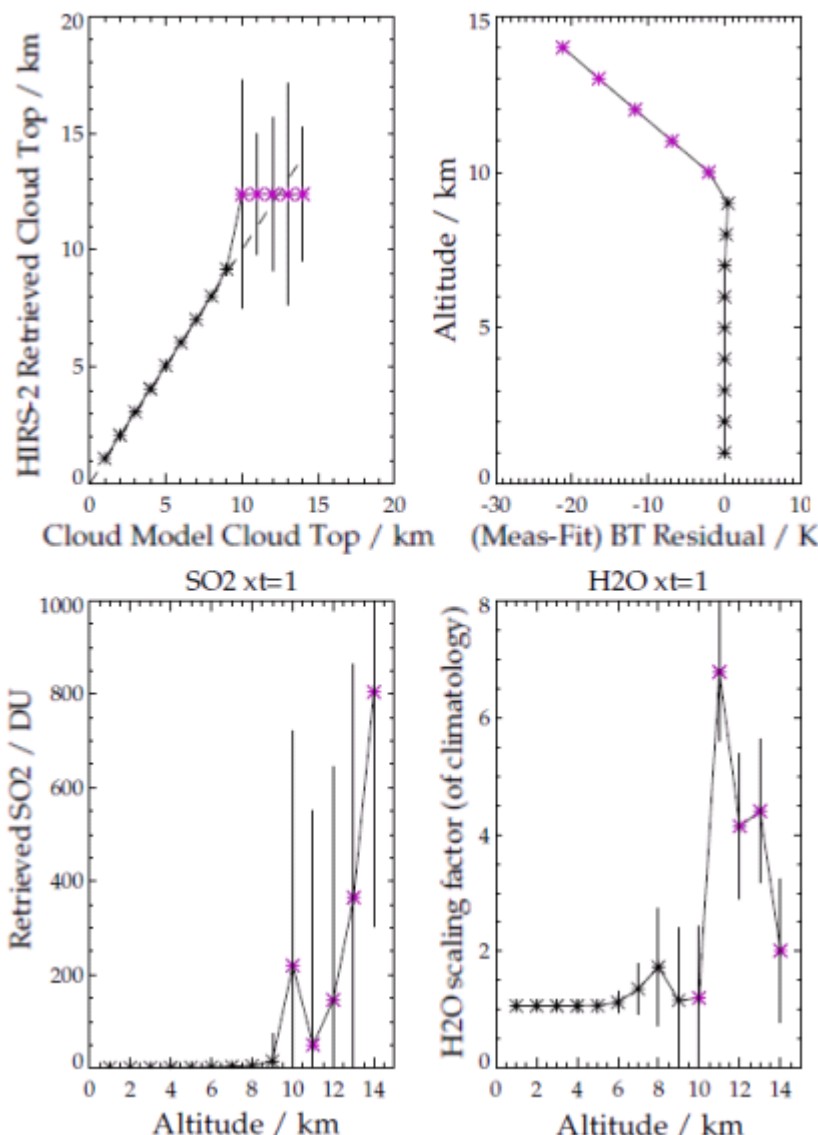

Figure 5. The top left plot shows retrieved cloud top height as a function of 'true' cloud top height as simulated by the cloud model. Black symbols indicate that the retrieval converged and purple indicates that it did not. The top right plot is of the fit residual (measurement minus fit) in the 11.1 μm channel. The bottom left plot shows the retrieved $SO_2$ as a function of the cloud top height in the cloud model, and the bottom right the equivalent for the water vapour scaling factor.

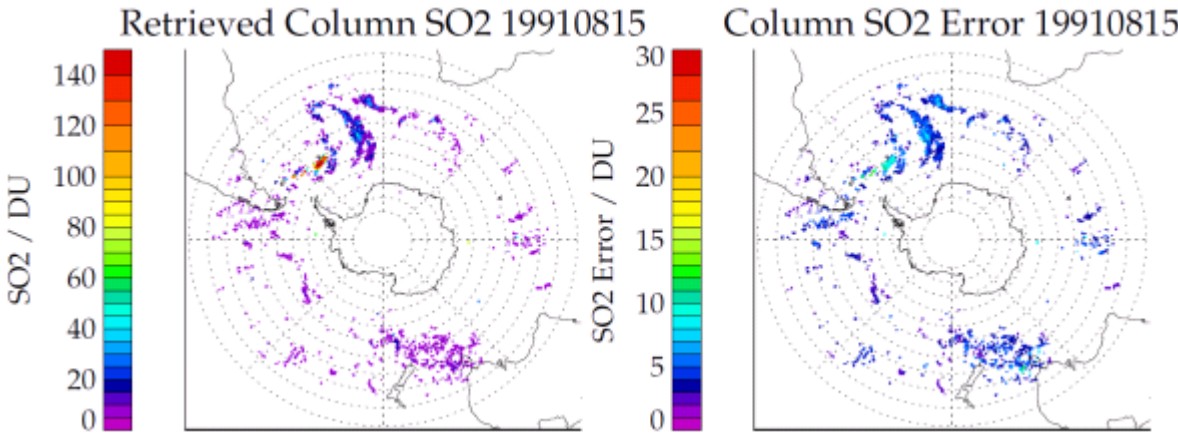

Figure 6: Retrieved SO$_2$ columns for 15th August 1991, and retrieved error for orbits that day. Erupted SO$_2$ from the start of the eruptive phase (from 8th August 1991) is evident ahead of the larger plume emitted on 15th August. Data are screened at the 2-sigma level (5.4 DU).

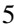

**Figure 7: The top left and top right show the retrieved water vapour scaling factor and its error from the column retrieval. The bottom left and right the equivalent for the retrieved cloud top height.**

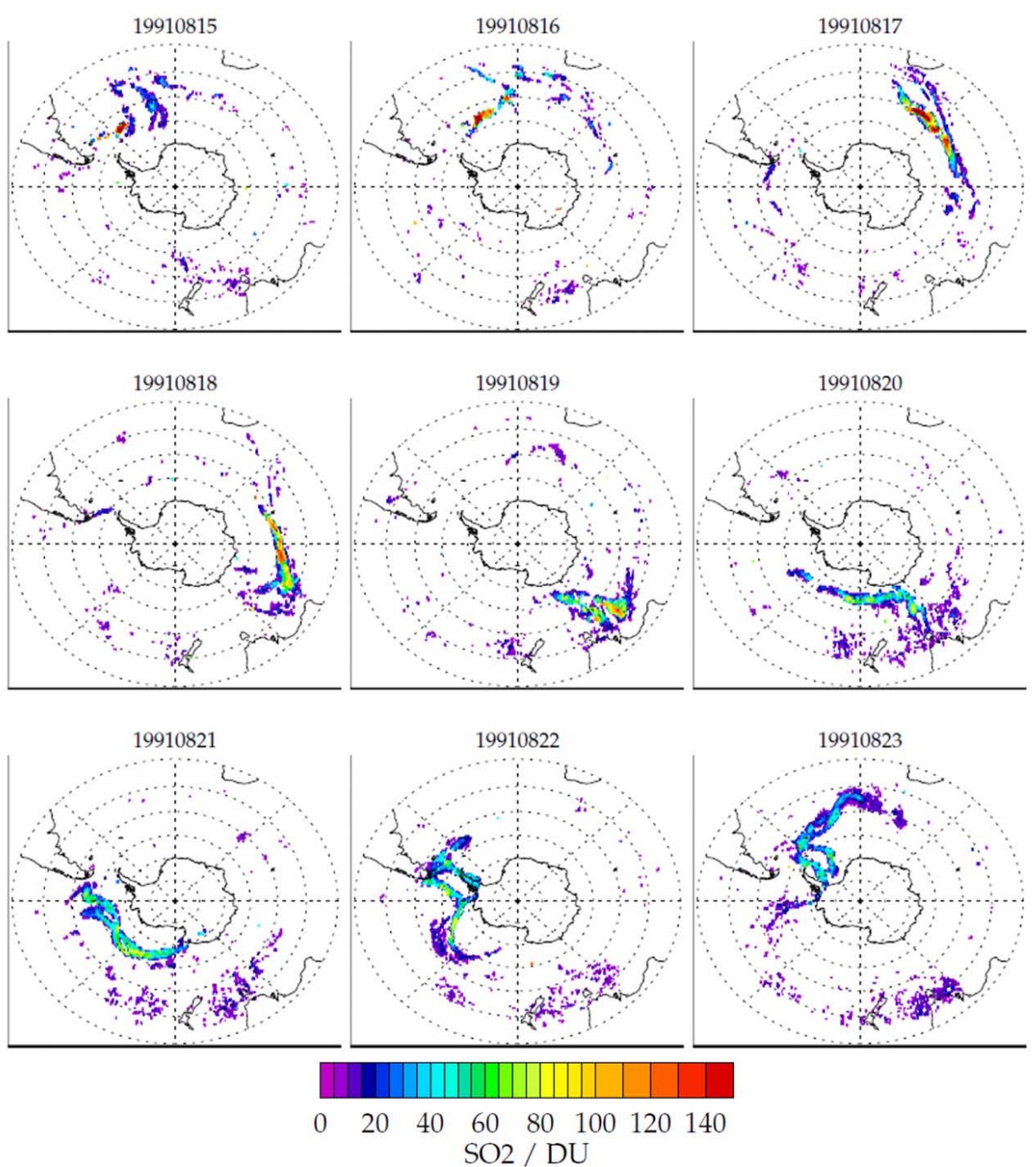

**Figure 8: Progression of main erupted plume from 15th August 1991, using all orbits (day and night) from HIRS/2 NOAA11. The eruption began with smaller amounts emitted from 8th August, which are apparent on 15th and disassociated from the main plume. The plume's transport between observations is evident, particularly from 21st August, where it is captured multiple times by multiple swaths. Data have been screened at the 3-sigma level (8.1 DU) for clarity of the main plume.**

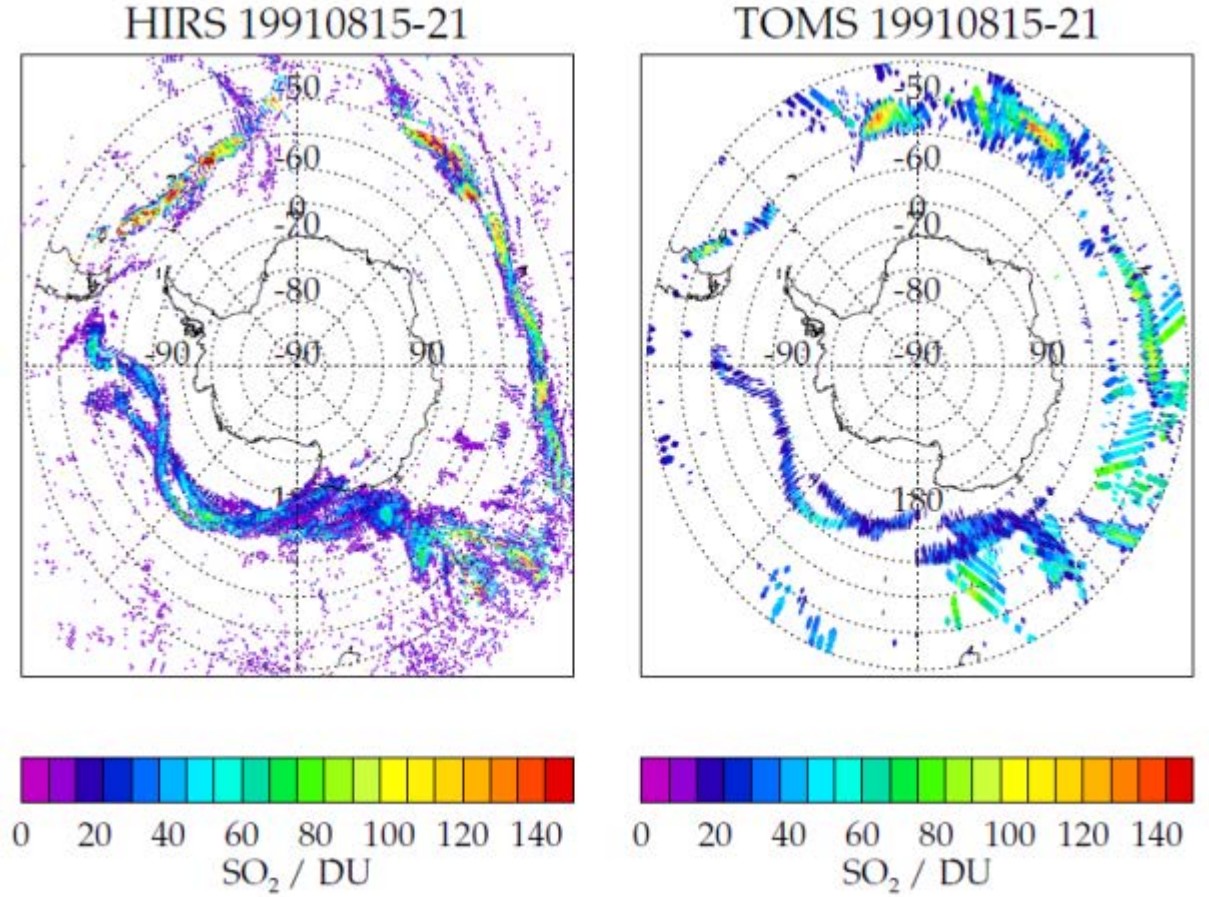

**Figure 9: Seven day composite of retrieved SO$_2$ from 15-21st August 1991. For clarity in comparison, TOMS data are screened to have a minimum value of 15 DU and HIRS/2 data uses 3 sigma (7.1 DU)**

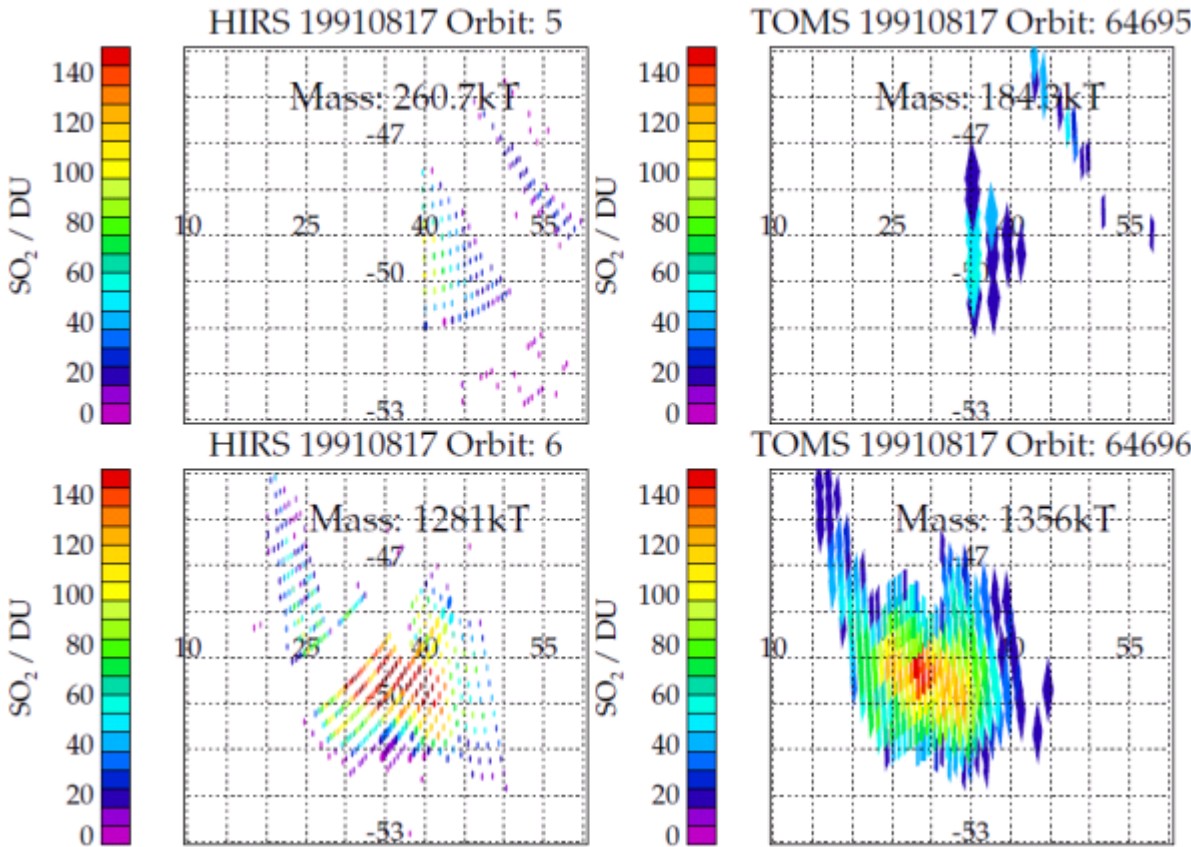

**Figure 10: The main Hudson plume on 17$^{th}$ August 1991 as observed in orbits 5 and 6 by HIRS/2 and 64695 and 64696 by TOMS, two days after the main paroxysmal eruption that occurred on 15$^{th}$ August. Four scan lines in the HIRS/2 panels are missing due to routine a calibration phase in which no data are provided. HIRS and TOMS data are both screened at the quality level of 2 - sigma level (5.4. DU and 15 DU respectively).**

**Table 1. Instruments (many of which were flown aboard several different platforms which are not listed) that have been used to measure volcanic SO$_2$ in the atmosphere.**

| Instrument name | Viewing geometry, spectral region | Period of operation | Relevant reference |
|---|---|---|---|
| TOMS, TOMS-like instruments (e.g. SBUV/2) | Nadir, UV | 1979+ | Krueger (1983), Kerr et al. (1980); Krueger et al., (1995, 2007); Guo et al. (2004) |
| HIRS/2 | Nadir, IR | 1979+ | Prata et al., 2003, this work. |

| | | | |
|---|---|---|---|
| MLS | Limb, IR | 1991+ | Read et al., (1993, 2009) |
| GOME, GOME-2 | Nadir, UV-vis | 1995+ | Eisinger & Burrows (1998); Khokhar et al. (2005); Nowlan et al. (2011); Rix et al. (2011) |
| ASTER | Nadir, IR imager | 1999+ | Pieri & Abrams (2004); Campion et al., (2010) |
| MODIS | Nadir, IR Imager | 1999+ | Watson et al., (2004) |
| SCIAMACHY | Nadir/Limb, UV-vis | 2002-2012 | Bovensmann et al., (1999); Gottwald et al., (2006); Lee et al., (2008) |
| MIPAS | Limb, IR FTS | 2002-2012 | Hoepfner et al., (2015) |
| AIRS | Nadir, IR Spectrometer | 2002+ | Carn et al., (2005); Chahine et al., (2006); Prata & Bernado (2007); Prata et al. (2010) |
| TES | Nadir, IR FTS | 2004+ | (Coheur et al. (2005); Clerbaux et al. (2005, 2008)) |
| SEVIRI | GEO, vis/NIR/IR imager | 2005+ | Prata & Kerkmann (2007); Thomas & Prata (2011) |
| IASI | Nadir, IR FTS | 2006+ | Karagulian et al. (2010) |
| OMI | Nadir, UV | 2006+ | Krotkov et al. (2010); Yang et al. (2007) |
| Suomi NPP OMPS | Nadir/Limb, UV | 2011+ | Yang et al., (2013) |
| TROPOMI | Nadir spectrometer UV/vis | 2017+ | Theys et al., (2016) |

**Table 2. HIRS/2 Instrument Parameters**

Instrument Parameter

| Cross-track scan | ± 49.5 ° (± 1125 km) nadir |
|---|---|
| Number of steps | 56 |
| Optical Field Of View | 1.25 ° |
| Step angle | 1.8 ° |
| Ground resolution IFOV (nadir) | 17.4 km diameter |
| Ground resolution IFOV (end of scan) | 58.5 km by 29.9 km |
| Distance between IFOV's | 42 km along track and nadir |

**Table 3. Total erupted $SO_2$ rounded estimates for Cerro Hudson**

| Eruptive Phase | TOMS $SO_2$[1] | TOMS $SO_2$[2] | HIRS/2 Prata fit[3] | HIRS/2 OE[4] |
|---|---|---|---|---|
| 8-9th August | 700 kT | - | 300 kT | 500 ± 150 kT |
| 12 August | 600 kT | - | 400 kT | 300 ± 90 kT |
| 15 August | 2700 kT | 2000 kT | 1200 kT | 1500 ± 400 kT |

[1]Constantine et al. (2000), with errors estimated to be circa 30 %.

[2]This work, based on updated TOMS algorithm, for total mass as observed on 16th August (as region poorly observed on 15th) with consideration of pixel overlap within orbit

[3]After Prata et al. (2003) but data reproduced and sampled as OE HIRS/2 product is herein.

[4]This work, with retrievd error.