# Peer review of "Retrieval of volcanic SO2 from HIRS/2 using optimal estimation"

_Atmospheric Measurement Techniques, 2017_

## Referee Comment (RC1) · Anonymous Referee #1 · 7 Mar 2017

The manuscript presents a new scheme for the retrieval of atmospheric SO2 total column amounts after volcanic eruptions from HIRS/2 observations. An optimal estimation retrieval using radiances from three HIRS/2 channels in the mid-infrared region is presented. Retrieved parameters are cloud top and the total column amounts of water vapour and SO2. Major error sources, which are identified by synthetic observations, are cloud/ash interference and the assumptions on the altitude and vertical extend of the SO2 plume. Through simulations it is further demonstrated that the new scheme is superior compared to simple brightness difference methods. The method has been applied to the case of the eruption of the Cerro Hudson volcano in 1991.

This is an important piece of work since it presents an improved retrieval scheme to obtain SO2 from TOVS measurements and, thus, opens the possibility to obtain climatological time series of this important trace species. After some modifications/extensions

as detailed below, I strongly support its publication in AMT.

General comments:

The optimal estimation scheme is used but not explained nor referenced. I would propose to add a paragraph with the main formulas (adapted to the actual retrieval problem) and add the main references. It would be very helpful to add a table or some graph summarizing the major error terms which have been investigated and how those are handled (some explicitly, some implicitly as part of the measurement error).

Specific comments:

P1L20: 'detection method': The method presented here is more than pure 'detection' – it's quantification.

P2L6: could you specify more precisely the channel boundaries for the different HIRS/2 instruments. How much does this affect the retrieval scheme?

P4L24: It would help the reader if a higher resolved spectrum of the single major contributors to the radiance could be provided, overlaid by the channel-boundaries. Are there any other gases contributing in each channel?

P5L19, 'up to 300 DU': Could you give examples from literature how this number covers the upper limit of volcanic eruptions. In addition, the spectral plots (see comment P4L24) should include lines of $SO_2$ for different column amounts.

P5L24 'see Fig. 1': Fig. 1 has not been described up to this point, but only later in the text. Further, it does not show the cost 'up to the training limit' of 300 DU, but only to 200DU.

P5L30 'calculated numerically': Could you explain this more in detail. Are the analytic Jacobians used at all?

P5L31, 'manually': What does this mean? How large are the limits for $H_2O$, $SO_2$? How does it work when it is mentioned 'The weighting functions are allowed to make linear

extrapolations. . .'? Is this only valid for the last iteration step?

P6L25, 'The estimate accounts for inaccuracies that arise due to modelling the atmosphere at reduced spectral resolution, limited vertical resolution, inclusion of non-retrieved trace gases at a climatological level or their preclusion entirely, relative to a reference case.': What is meant with 'limited vertical resolution' and 'modelling .. at reduced spectral resolution'? How has this error been derived (line-by-line compared to band model)? How strong does this error depend on the atmospheric situation?

P7L9, '100 DU': Could you put this number in perspective of the typical maximum column amounts e.g. after Pinatubo?

P7L16: How are the a-priori errors of the state vector element for water vapour set? Is this error only considered in the measurement space? Further, could you explain how the off-diagonal elements of the a-priori covariance matrix are set.

P8L1, Chapter 3.1, and Fig.1: Regarding the error bars shown in Fig. 1: Could you summarize which errors they contain? Have these errors been incorporated in the synthetic observations? (I assume no or only partially, otherwise the retrieval results should somehow scatter around these errors). Further, do the error bars represent 1 or 2-sigma values? The maximum value tested here seems to be 150 DU. Could you extend this range? You should also show, at which values the method fails and at which column amount of SO2 the channel signal becomes saturated.

P8L25, 'Measurements were simulated for a plume at a range of altitudes from 8-18 km': However the caption in Fig. 2 says vice-versa: 'A 2 km thick triangular profile centred at 12 km is used to simulate measurements. The profile is then used in a retrieval with the retrieved height assigned to a range of altitudes.' Could you tell in which way the test retrievals have really been performed?

P8L28: In this paragraph it is only referred to the Figure, however the results are not described. Please give also in the text at least some quantification of the resulting

errors.

P9L1-3: It is not clear what is different here compared from the paragraph before.

P9L6-13: Also here in the text some numbers (%error) should be mentioned. Further, could you explain, why there is such a large difference between the errors when the plume thickness is over- versus underestimated. Would this result not speak for application of a rather sharp profile in the retrieval to minimize the errors?

P9L29: 'water vapour clouds' should perhaps read 'liquid water clouds' ?

P9L29, 'above 5 km': However in Fig. 4 the retrieval seems to be OK up to 8-9 km. Can you give an explanation why the retrieval has problems to fit cloud heights above a certain altitude. How much does it depend on the atmospheric situation (tropics vs mid/high latitudes)?

P10, chapter 3.4: What is the upper limit of the retrieved SO2 (e.g. due to saturation effects)? This could also affect the total mass calculation in very dense plumes. Also some information about the convergence criteria of the retrieval and how many iterations are necessary are missing.

P12L28: Could you state which SO2 altitude profile has been used for the case study and how it has been derived. Is the resulting SO2-altitude error included in the column errors in Fig. 5?

P14L5: To make this calculations more clear and give the reader a better feeling for the derived e-folding times and its possible uncertainties it would be necessary to plot derived daily masses after 17th August and show the fitted exponential decay line.

P14K19, '2300 +- 600 kT': How has the error of 600 kT been calculated?

P14L22: Please give also the total masses (including errors) of TOMS, Carn et al., 2016 and Prata et al., 2003.

Technical comments:

P1L31, 'The TOVS instrument': But in the sentence before it is explained as a suite of instruments.

P1L32, 'TIROS': Is written 'TIrOS' in L28.

P4L21, 'Table 1': Shouldn't this read 'Table 2'?

P11L7: '(Constantine et al. 2000)' -> '(Constantine et al., 2000)'

P12L14: 'verses' -> 'versus'

P13L4: '317 channel' -> '317 nm channel'

P13L5: '340 to' -> 'channel at 340 nm to'

P14L3: Should the formula not read: $N(t) = N_0 \exp(-\lambda t)$ ?

P15L15: 'satellites' -> 'satellite'

P23Fig3: '11.5, 12, 13 km' should read '11.5, 12, 12.5 km'

---

## Referee Comment (RC2) · Anonymous Referee #2 · 8 Mar 2017

General comments

The authors present a new algorithm for the retrieval of volcanic SO2 total column amount from the HIRS/2 instrument. This paper is well structured and convincingly demonstrates the added value of adding HIRS/2 to the series of instruments used for the retrieval of SO2.

Indeed, as stated in the paper, long-term and systematic monitoring of volcanic SO2 is relevant in relation to climates issues and knowledge on plume evolution. I enjoyed reading the paper and would certainly like to see it published in AMT, after taking into account the remarks below.

1 - Introduction Well written. Clearly indicates the relevance of performing SO2 retrievals on HIRS/2 measurements, an instrument originally not devised for that purpose. Although a method to derive SO2 from HIRS already exists (the Prata fit method), the paper indicates the shortcoming of that method and outline how the retrieval could be improved my taking multiple HIRS channels into account by using an OE scheme.

2 - Methodology Section 2.1 introduces the HIRS channels to be used in the OE retrieval, as well as the applied RTM, RTTOV.

P4 L17. Pleas mention briefly why NOAA11 was selected.

P5 L19-24: It may be beneficial to train the model for amount larger than 300 DU, as (much) higher values occasionally occur in the most powerful eruptions (e.g. Nabro in 2011). Is there a specific reason to limit the procedure to 300DU? The reference to Figure 1 seems premature, as this figure is discussed only later in the paper and shows total SO2 amounts up to 200 DU only. I suggest not mentioning the figure or to explicitly state that this figure is to be discussed in more detail later in the text.

Section 2.2: P6 L10-12: The later assessment of retrieval sensitivity to uncertainties in plume altitude and thickness is introduced here. The vertical extend of the plume is said to be derived from ancillary information. I think it would be good to state that all parameters involved here are effective values, certainly when using a pre-described triangular profile shape. For example, in reality the SO2 profile may show multiple peaks at different altitudes. Knowing this, the assumption of a triangular shape is as good as any other.

3. Error study. Overall a clear, to the point chapter.

Section 3.2.1 took me a bit longer to understand. P8 L25-28: The texts states that retrievals are performed on simulated spectra, with a fixes plume altitude of 12 km assumed in the retrieval. However , Figure 2 suggests that the peak altitude is fixed in the RFM simulations. Which is correct?

P9 L1-3: What is meant here? Where retrievals also performed using the RFM as forward model, with the conclusion that it performed less well for plumes > 17 km than

[Figure]

RTTOV?

Section 3.2.2 P9 L9-10. Any idea as to why an underestimation of the plume thickness has significantly less impact on the result than an overestimate? Would this depend on the peak altitude of the plume.,bringing it in another temperature/water vapour domain?

Section 3.3: The seem to be some little inconsistencies here. The section states that care should be taken with clouds above 5-6 km. A threshold of 5 km is used further on in the paper, whereas the abstract mentions 6 km. Yet, figure 4 suggests that one can go as far as 9 km without any significant problems.

4. Case study.

Please add a few words on why this particular eruption was selected to demonstrate the new algorithm. Also, this eruption is compared to Kasatochi in the abstract, something that you may want to repeat here.

The assumed plume altitude and thickness of the plume is not mentioned in the text. From the description of previous studies (putting the Cerro Hudson aerosols at 11-13 km) I assume that you used the same 12 km plume as used for Section 3, but this is not clear.

5. Discussion Despite the remaining uncertainties in the new algorithm, the authors manage to demonstrate the added value of the new system in comparison to previous methods. It would be nice to see a short summary here of what drawbacks of the Prata methos have been resolved by the new OE schemes and which issues remain, such as the dependence on plume altitude information.

I very much liked the clear understanding by the authors that the presented work can be seen as a mere first step toward extending and improving the long-term data series of volcanic SO2 measurements from satellite. I certainly hope that (part of ) the proposed future work will be realized.

Cosmetics:

P2 L28: TiROS –> TIROS; Or keep TiROS and used this spelling consistently through-out the paper.

P13 L1-5: The wavelength unit (nm) is missing a few times.

P14 Eq 1: minus symbol missing in the exponent.

P23: Caption: 11.5, 12, 13 –> 11.5, 12., 12.5

---

## Author Comment (AC1) · 15 May 2017

article graphicx

[Figure]

**Authors' Response to Reviewer 2**

May 15, 2017

**Referee comments in italics.**

*General comments*

*The authors present a new algorithm for the retrieval of volcanic $SO_2$ total column amount from the HIRS/2 instrument. This paper is well structured and convincingly demonstrates the added value of adding HIRS/2 to the series of instruments used for the retrieval of $SO_2$. Indeed, as stated in the paper, long-term and systematic monitoring of volcanic $SO_2$ is relevant in relation to climates issues and knowledge on plume evolution. I enjoyed reading the paper and would certainly like to see it published in AMT, after taking into account the remarks below.*

*1 - Introduction Well written. Clearly indicates the relevance of performing $SO_2$ retrievals on HIRS/2 measurements, an instrument originally not devised for that purpose. Although a method to derive $SO_2$ from HIRS already exists (the Prata fit method), the paper indicates the shortcoming of that method and outline how the retrieval could be improved my taking multiple HIRS channels into account by using an OE scheme.*

*2 - Methodology Section 2.1 introduces the HIRS channels to be used in the OE retrieval, as well as the applied RTM, RTTOV.*

*P4 L17. Pleas mention briefly why NOAA11 was selected.*

NOAA-9, 10, 11 and 12 were in orbit and operational at the time of the Cerro Hudson eruption. NOAA-11 was selected to pilot the technique principally because of its simple channel configuration. Using the 8.6 micron channel, which is potentially sensitive to both ash and $SO_2$, would be considered to be an extension to this work since it would require the forward model to additionally be able to simulate ash, and proof of concept with just a single channel sensitive to $SO_2$ was the first goal. Furthermore, NOAA11 benefits from an extra window channel (which on the other instruments is the 8.6 micron channel) that can be used for offline detection by means of BT difference or ratio flags should further diagnostics be required. This use was explored but is considered to be beyond the scope of the work, since the alternative and more reliable simulations using a cloud and aerosol model were perused to investigate the limitations of the retrieval under cloudy or ash filled FOVs.
We have modified the text as follows:
"This instrument was selected to demonstrate the capability of this version of the instrument with only one channel that is sensitive to $SO_2$ and two window channels that have some potential to be used to flag cloud and under some circumstances ash if required."

*P5 L19-24: It may be beneficial to train the model for amount larger than 300 DU, as (much) higher values occasionally occur in the most powerful eruptions (e.g. Nabro in 2011). Is there a specific reason to limit the procedure to 300DU? The reference to Figure 1 seems premature, as this figure is discussed only later in the paper and shows total $SO_2$ amounts up to 200 DU only. I suggest not mentioning the figure or to explicitly state that this figure is to be discussed in more detail later in the*

*text.*

It would be beneficial to train the model for higher amounts than 300DU to accommodate the larger and more intense eruptions. The limit was chosen in this case to be appropriate for the case study eruption, where a priori knowledge existed (e.g. from TOMS) to suggest that in nearly all instances this would be sufficient. The training limit is very important due to the way in which RTTOV calculates layer transmittances for gases because some species require higher order terms in their predictor coefficients that are challenging to characterise. To train the model for higher $SO_2$ column amounts is probably possible, but would be the subject of further, future work as non-linearities in the behaviour of RTTOV at very high $SO_2$ loadings, sensitivity to profile shape and saturation effects would all have to be adequately examined.

In response to a comment from Reviewer 1, a new figure has been added to the manuscript as Figure 1, showing spectral transmittances for water vapour and two amounts of $SO_2$ (1DU and 300 DU) around the 7.3 micron channel. The reference to figure what is now figure 2 discussed her ehas been amended to reflect that it will be discussed in detail later on.

*Section 2.2: P6 L10-12: The later assessment of retrieval sensitivity to uncertainties in plume altitude and thickness is introduced here. The vertical extend of the plume is said to be derived from ancillary information. I think it would be good to state that all parameters involved here are effective values, certainly when using a pre-described triangular profile shape. For example, in reality the $SO_2$ profile may show multiple peaks at different altitudes. Knowing this, the assumption of a triangular shape is as good as any other.*

A broadly triangular profile may be considered a better representation than any other for a short eruption where material could be expected to gather at a height of

neutral buoyancy in the stratosphere where vertical sheer in the advection profile is more limited than in the troposphere over a short altitude range.

In tandem with a point from Reviewer 1, the text has been modified to elaborate on the profile used and the motivation for its selection. It is very probable that multiple peaks in the $SO_2$ profile may exist. This wasn't explicitly tested. Specifically in this case however, the profile used is considered to be a reasonable representation of the plume observed in the case study. As figure 3 demonstrates, there is some sensitivity to the thickness of the modelled plume – modelling it to be too thick introduces more error than under-estimating its thickness in the case tested here, although this is small compared to errors in height assignment. Often (e.g. for IASI and GOME-2) retrievals are performed assuming the material is at 3 altitudes, and the result which best fits the measurements is generally considered to be the 'best', but some human judgement (often based on ancillary information) is also required when looking at results for specific eruptions from these instruments or when considering total erupted mass. There is more information for the retrieval of altitude with spectrometers but when this is done the error on the amount of SO2 retrieved increases very substantially. In summary, an effective profile is often the best that can be done in the absence of any other information, but there may be some errors associated from getting it wrong – some of which needed to be explored here. As such, every effort must be made to make the profile and height as realistic as possible. More work was done than has been stated to identify the plume altitude from Hudson, and some of it bears repeating to explain the origin of the $SO_2$ profile used in the model, before it is stated. This is discussed in response to a point below, under Case Study.

That being said, we appreciate the point the reviewer is making here. Section 2.2. has had the following added to the start:
"In the absence of any further information, an effective $SO_2$ profile must be represented in the forward model."

*3. Error study. Overall a clear, to the point chapter.*

*Section 3.2.1 took me a bit longer to understand. P8 L25-28: The texts states that retrievals are performed on simulated spectra, with a fixes plume altitude of 12 km assumed in the retrieval. However , Figure 2 suggests that the peak altitude is fixed in the RFM simulations. Which is correct?*

This section now reads (referring to what is now Figure 3):
"Measurements were simulated for a plume at a range of altitudes from 8-18 km. Figure 3 shows the impact on the retrieved $SO_2$ column at a specified, fixed altitude of 12 km as a fraction of the true column at these altitudes. Errors range from typically $\pm$0-30 % for most column amounts up to 100 DU an increase for larger amounts, and for particular altitudes. While the specific error may be state dependent (upon meteorological conditions, specifically the water vapour profile), these simulations do give a general indication as to the magnitude of error that can result from incorrect height assignment of the volcanic plume in the forward model. This is the largest source of error in the OE column retrieval (and the Prata-fit method) and is made more challenging because there is a dependency of the error on column amount. Since height assignment errors cannot be known such simulations can at least give a general indication of potential uncertainty of retrieved amounts, depending on the quality of information available regarding altitude of volcanic $SO_2$. It is clear therefore that good prior knowledge of the $SO_2$ plume altitude is necessary for accurate retrieval or fit of $SO_2$ column amounts from HIRS/2."

The caption for what is now figure 3 has been amended to read:
"A measurement was simulated for a volcanic plume of triangular profile centred at a range of altitudes, for a range of total column amounts. A retrieval is then performed where the plume is assumed to be at 12 km. The fractional difference, or error, is

plotted."

*P9 L1-3: What is meant here? Where retrievals also performed using the RFM as forward model, with the conclusion that it performed less well for plumes > 17 km than RTTOV?*

The text has been modified to make its purpose more clear:
"The performance of the column fit was also directly assessed against a line-by-line model (RFM) for plume altitudes from 8 to 18 km (where the plume height assignment used in the retrieval was the same as that used in the measurement simulated by the RFM) and it was found that...". Specifically, it refers to a test of precision/accuracy of RTTOV vs RFM forward models.

*Section 3.2.2 P9 L9-10. Any idea as to why an underestimation of the plume thickness has significantly less impact on the result than an overestimate? Would this depend on the peak altitude of the plume, bringing it in another temperature/water vapour domain?*

We are not certain, but as suggested it probably has something to do with where it is in the atmosphere and how this relates to the water vapour. It does suggest that so long as the plume is located (and modelled) above most of the atmospheric water vapour, it is better to underestimate the plume thickness or else use a very narrow profile in similar cases.
The text has been modified as follows:
"The retrieval simulations suggest that errors are larger when the plume thickness is overestimated (typically 13 %), with only small inaccuracies introduced when the plume thickness is under-estimated (less than 2 %). The modelled cloud top height was 3 km in all cases. It is therefore possible that an underestimate of plume thickness would result in smaller errors."

*Section 3.3: The seem to be some little inconsistencies here. The section states that care should be taken with clouds above 5-6 km. A threshold of 5 km is used further on in the paper, whereas the abstract mentions 6 km. Yet, figure 4 suggests that one can go as far as 9 km without any significant problems.*

We thank the reviewer for pointing out these inconsistencies. The pertinent panel in figure 4 is the bottom right, which shows that deviations (errors) in retrieved water vapour column begin when a cloud is at 6 km. Poor fitting of water vapour leads to errors in the retrieval of $SO_2$, because the 7.3 micron channel is sensitive to both water vapour and $SO_2$. A cloud at 5 km shows no perceptible deviation, and it is likely that the threshold is somewhere in between, which is the origin of the 5-6 km warning. In the abstract 6 km was stated as this will definitely contribute an error to the retrieval of $SO_2$. For clarity this has been corrected to 5 km since it is stated as "...above.." the given level.

*4. Case study. Please add a few words on why this particular eruption was selected to demonstrate the new algorithm. Also, this eruption is compared to Kasatochi in the abstract, something that you may want to repeat here.*

The following text has been added:
"In this sense, as well as being a non-equatorial eruption, it has similarities to the 2008 Kasatochi eruption in the Northern hemisphere. It is selected here as a case study because it was a relatively large eruption that has not been studied exhaustively, and a very good example of an eruption in recent satellite history which only TOMS observed with any significance, that can benefit from application of this technique."

*The assumed plume altitude and thickness of the plume is not mentioned in the text. From the description of previous studies (putting the Cerro Hudson aerosols at 11-13 km) I assume that you used the same 12 km plume as used for Section 3, but*

*this is not clear.*
This has now been clearly stated, as well as the reasons for it:
"In addition, contemporary lidar measurements of the Hudson plume were made at the CSIRO (Commonwealth Scientificand Industrial Research Organisation) Division of Atmospheric Research, at Melbourne, Australia (38 S, 145 E) (Young et al., 1992, Barton et al. 1992). These measurements are sensitive to ash, sulphate aerosol and meteorological (water) cloud. The backscatter profiles tend to indicate peaks at around and above 20 km, and frequently at 10-13 km. The higher peak is attributed to aerosol from the Pinatubo eruption. Young et al. (1992) interpret the majority of observations that are thought to include Hudson material as the feature at 12 km in October, with variable cirrus at 10 km. It is reported by the authors that the plume was observed consistently from 28th August until December 1991 between 10 and 13 km, with a decreasing scattering ratio. The relative proportions that contribute to the backscatter measured are expected to be dominated by ash in the first few weeks after the eruption. Little ash is expected to be present after a month beyond the eruption, but by this time the vast majority of the $SO_2$ will have oxidised into aerosol. Whilst lidar is not sensitive to the presence of gaseous $SO_2$ inferences can be drawn from the height of the aerosol it eventually becomes. In this case the lidar information is considered to be considered a starting point as a guide for estimating the cloud height of the $SO_2$, to be considered in the context of other information."
Additionally:
"Using all of this information, the Hudson plume is modelled as a triangular peaked profile with a baseline of 2 km between 11 and 13 km, peaking at 12 km."

*5. Discussion*
*Despite the remaining uncertainties in the new algorithm, the authors manage to demonstrate the added value of the new system in comparison to previous methods. It would be nice to see a short summary here of what drawbacks of the Prata methods have been resolved by the new OE schemes and which issues remain, such as the*

*dependence on plume altitude information.*

The list has been slightly expanded upon to include some more points:
"They include a quantified error on individual pixel retrieved values, latitudinal variation in accuracy, diagnostic indicators of the retrieval performance and goodness-of-fit and treatment of cloud and water vapour consistent to the retrieval of $SO_2$. When summing mass over a large number of pixels, the precision that these afford becomes increasingly important. Issues that remain are those endemic to ill posed problems where there is only one piece of information on $SO_2$ available and only limited information about the height or shape of the profile of a volcanic plume. It is conceivable that further progress might be made by using HIRS/2 aboard NOAA10 and 12 with the addition of the 8.6 $\mu$m channel in ash-free pixels."

*I very much liked the clear understanding by the authors that the presented work can be seen as a mere first step toward extending and improving the long-term data series of volcanic $SO_2$ measurements from satellite. I certainly hope that (part of) the proposed future work will be realized.*

We thank the reviewer for their support of what we consider to be useful work. This work has so far not had the benefit of any direct funding, but it is hoped that it may contribute to a future case for support for funding. We welcome any opportunities for collaboration.

*Cosmetics:*
*P2 L28: TiROS to TIROS; Or keep TiROS and used this spelling consistently throughout the paper.*
These have been unified to read TIrOS.

*P13 L1-5: The wavelength unit (nm) is missing a few times.*

This has been corrected.

*P14 Eq 1: minus symbol missing in the exponent.*
This has been corrected.
*P23: Caption: 11.5, 12, 13 to 11.5, 12., 12.5*
This has been corrected.

---

## Author Comment (AC2) · 16 May 2017

article graphicx

**Authors' Response to Reviewer 1**

May 16, 2017

**Referee comments in italics.**

We would like to thank Reviewer 1 for his detailed comments which have improved the paper. *The manuscript presents a new scheme for the retrieval of atmospheric SO$_2$ total column amounts after volcanic eruptions from HIRS/2 observations. An optimal estimation retrieval using radiances from three HIRS/2 channels in the mid-infrared region is presented. Retrieved parameters are cloud top and the total column amounts of water vapour and SO$_2$. Major error sources, which are identified by synthetic observations, are cloud/ash interference and the assumptions on the altitude and vertical extend of the SO$_2$ plume. Through simulations it is further demonstrated that the new scheme is superior compared to simple brightness difference methods. The method has been applied to the case of the eruption of the Cerro Hudson volcano in 1991. This is an important piece of work since it presents an improved retrieval scheme to obtain SO$_2$ from TOVS measurements and, thus, opens the possibility to obtain climatological time series of this important trace species. After some modifications/extensions as detailed below, I strongly support its publication in AMT.*

*General comments:*

*The optimal estimation scheme is used but not explained nor referenced. I would propose to add a paragraph with the main formulas (adapted to the actual retrieval problem) and add the main references. It would be very helpful to add a table or some graph summarizing the major error terms which have been investigated and how those are handled (some explicitly, some implicitly as part of the measurement error).*

We thank the Reviewer in particular for pointing out the missing identification of the optimal estimation scheme. This is a clear oversight and some indication of inverse method used is important to be included in the paper, even if it is only a mathematical tool. We have now referred in the text to Rodgers (2000) generally and Miles et al., (2014) specifically, the latter of which used identical retrieval methodology in terms of the inverse method and cost minimisation part of the retrieval used with the forward model. In that text the methodology is explained quite exhaustively.

The issue of the handing of errors can be elucidated, indeed it could be made clearer that some sources of error can be handled by the retrieval (presence of cloud/ash, $SO_2/H_2O$ covariance, measurement noise and an attempt at FM error), and others can only be explored to obtain a general indication of the sort of confidence that may be placed on the results (such as height and plume thickness uncertainty). Specific comments:

*P1L20: 'detection method': The method presented here is more than pure 'detection' – it's quantification.*

Text has been changed to "..detection and quantification method...".

*P2L6: could you specify more precisely the channel boundaries for the different HIRS/2 instruments. How much does this affect the retrieval scheme?*

The following text and references have been added:
"Retrievals are obtained using the Levernburg-Marquart minimisation method after Rodgers (2000), and the full optimal estimation scheme used here is described in detail in Miles et al., (2015)."

Rodgers, C., "Inverse methods for atmospheric sounding: theory and practice", 1 edn, World Scientific, 2000.

Miles, G. M., Siddans, R., Kerridge, B. J., Latter, B. G., and Richards, N. A. D.: Tropospheric ozone and ozone profiles retrieved from GOME-2 and their validation, Atmos. Meas. Tech., 8, 385-398, doi:10.5194/amt-8-385-2015, 2015.

The following text has been added/amended in the discussion on error study, section 3:

"There are some sources of error that can be incorporated and dealt with by the retrieval. These include measurement noise, the presence of cloud or ash, $SO_2/H_2O$ covariance and an estimate of forward model error discussed above. The main sources of error that cannot be adequately represented in the forward model are errors that impact ill-posed nadir $SO_2$ column retrievals in general. These are incorrect height assignment of the $SO_2$ plume, incorrect thickness in the plume represented in the forward model and particularly in the case of infrared measurements and sensitivity to the presence of cloud and/or water vapour. Their relative impacts vary and the sensitivity of the solution to them can be quantified using simulations. It should be noted that some of these errors (plume height and profile shape) cannot often be known at the time of retrieval, and as such the actual impact on the retrieval result also cannot be known. They are investigated here in order to give a general indication as to the potential error that can be associated with the results, to give a window of confidence. Others, such as the impact of cloud or ash on the retrieved $SO_2$ error can

be investigated for use in quality control."

*Specific comments:*
*P1L20: 'detection method': The method presented here is more than pure 'detection'*
*– it's quantification.*

Text has been changed to "..detection and quantification method. . .".

*P2L6: could you specify more precisely the channel boundaries for the different*
*HIRS/2 instruments. How much does this affect the retrieval scheme?*

There are 15 instruments on just the TiROS/NOAA platforms (1978-2005 before
MetOp and HIRS/4) each instrument had similar but slightly different channel configu-
rations but where channels are considered to be in common the central wavenumber
is similar. It would be a distraction and exhaustive to describe each of the channel
boundaries for each of the instruments, but it is sufficient in the authors' view to
mention that the channels can vary between platforms/instruments, but all broadly
have the three channels used in the retrieval. It has not been fully investigated for the
OE scheme but other HIRS/2 instruments have been used by the Prata fit method in
the literature.

This instrument is a broadband radiometer rather than a spectrometer, so small
differences in the central wavenumber of channels between instruments is not consid-
ered sufficient to appreciably alter or limit this approach between instruments for the
OE scheme. They were designed to respond to specific, principal absorbing species
relevant to the purpose of the instrument, which was to characterise temperature
profiles, water vapour, total ozone, cloud top pressure and surface reflectance, and as
such do not change much between instrument as they were designed to be similar.
The width of the channels is pertinent if there are multiple absorbing species within

the envelope of the instrument response function. All such species are required to be taken into account in the forward modelling of the atmosphere if they impact the measurement such that their absence would contribute to model error.

*P4L24: It would help the reader if a higher resolved spectrum of the single major contributors to the radiance could be provided, overlaid by the channel-boundaries. Are there any other gases contributing in each channel?*

Other gases do indeed contribute to the channels, but the predominant species (the absence of which would contribute appreciable error in modelled channel radiance) are included and modelled in RTTOV at a climatological value if they are not retrieved. The error of including at climatological value was investigated in great detail using the RFM as a way of estimating forward model error (in addition to the impact of modelling the atmosphere at the lower spectral resolution of RTTOV compared to the RFM). Other potential contributions to forward model error that can be estimated this way include spectral resolution of the forward model, including in the FM non-retrieved gases at their climatological value, excluding other gases which are known to exist in the real atmosphere and representing the vertical atmosphere at limited height resolution.

Error contributions deemed significant in the discussion of the RFM used to estimate forward model error (FME) are those that for a channel stand out from the others in terms of magnitude, and that exceed the noise equivalent brightness temperature difference for a given scene temperature. The absolute difference between a test case and the reference case is taken as the channel contribution of FME. The simulations were performed for all 19 of the HIRS/2 channels, irrespective of the fact that only three are used in the column retrieval. Extensive simulations with the RFM were performed that tested the sensitivity of the channel brightness temperatures to variation of the

elements listed above. The RFM was run with spectral resolution ranging from 0.001 cm−1 to 0.1 cm−1 to quantify the effects of a reduction in spectral resolution. This only appreciably impacted channels not used here. The change in simulated channel brightness temperature for gaseous species at their climatological level and 1 standard deviation from it were used to quantify the individual impacts of non-retrieved trace gas variability. This only really impacted the channel used to detect column ozone and not used in the retrieval. The RFM was also used to simulate the impact of including all of the minor species such as SF6 and F12-14 (anthropogenic halides). They showed that provided one accepted that their variability was low in the real atmosphere, there was very little sensitivity to them and they did not require inclusion in the background profile used in the forward model, but by this method their exclusion contributed quantitatively to the FME. All elements of forward model error were combined in quadrature. The forward model error as defined by calculations with the RFM is not definitively appropriate to a forward model based on RTTOV. It will contribute to the estimate of the total FME for the purposes of this method development in the absence of an equivalent term being evaluated for RTTOV (which is considerably more challenging to obtain), and broadly constitute a minimum envelope of FME for the HIRS/2 channels. This exercise has only been performed for HIRS/2 NOAA11, but since the channels in the retrieval are very similar, the FME contribution is not expected to change appreciably, but could be the subject of future work.

The above is mentioned but very succinctly in section 2.3 for the sake of brevity and concentrating on the channels used in the retrieval.

The following text has been added to section 2.1 to be more clear:
"Other atmospheric gases not retrieved but contribute appreciably to channel brightness temperature are represented in the forward model by a climatological value. The potential error that this can introduce is incorporated into the estimate of forward model error."

1 has been added to the text as the new Figure 1, in addition to the following text:

"Channel 11 from HIRS/2 aboard NOAA11, centred on 7.2 $\mu$m, is shown in Fig. 1. Also shown are simulated transmission spectra for water vapour (which this channel was designed to detect) and $SO_2$, for two column amounts. It demonstrates both that the channel and spectral feature coincide well, and for large column amounts of $SO_2$ the channel would be strongly affected."

It is not thought necessary to plot the other two channels, since they are a further water vapour channel and window channel and are not particularly illuminating.

The following has been added to the text:

"Further information about the principle absorbers of the other channels not used in the retrievals can be found in NOAA (1981)."

NOAA (1981), NOAA Technical Report NESS 83: Atmospheric Sounding User's Guide, Technical report, National Oceanic and Atmospheric Administration. NESS 83.

it P5L19, 'up to 300 DU': Could you give examples from literature how this number covers the upper limit of volcanic eruptions. In addition, the spectral plots (see comment P4L24) should include lines of $SO_2$ for different column amounts.

It is not stated that this number (300 DU) covers the upper limit of volcanic eruptions. The limit was chosen in this case to be appropriate for the case study eruption (and those smaller), where a priori knowledge existed (e.g. from TOMS, references given in text) to suggest that in nearly all instances this would be sufficient. This would be the case for the 2008 eruption of Kasatochi which was of a similar magnitude. The training limit is very important due to the way in which RTTOV calculates layer transmittances for gases because some species require higher order terms in their predictor coefficients that are challenging to characterise. To train the model for higher

[Figure]

New_fig_1.jpg

**Fig. 1.**

SO$_2$ column amounts is no doubt possible, but would be the subject of further, future work as non-linearities in the behaviour of the model at very high SO$_2$ loadings, sensitivity to profile shape and saturation effects would all have to be adequately examined.

In order to put column amounts discussed into context, the following text has been added:
"100 DU represents an SO$_2$ column from a large, explosive volcanic eruption. Pinatubo, for example, yielded column amounts of 350-500 DU (depending upon instrument) after 24 hours which reduced to 100 DU after 7 days (Carn et al., 2005). The OMI instrument (see Table 1) captured column amounts of around 200 DU after the 2008 eruption of Kasatochi (Prata et al., 2010)."
Spectral transmittances for two column amounts are now given in a new Figure 1.

*P5L24 'see Fig. 1': Fig. 1 has not been described up to this point, but only later in the text. Further, it does not show the cost 'up to the training limit' of 300 DU, but only to 200DU.*

This has been amended to reflect the actual scale used.

*P5L30 'calculated numerically': Could you explain this more in detail. Are the analytic Jacobians used at all?*

They are evaluated numerically by successive calls to the forward model for fractional perturbations of the state vector. The analytical Jacobians aren't used directly in the retrieval.
The text has been changed as follows: "As a result, these are evaluated numerically in the forward model by successive FM calls where each element of the state vector is fractionally perturbed in turn"

*P5L31, 'manually': What does this mean? How large are the limits for $H_2O$, $SO_2$? How does it work when it is mentioned 'The weighting functions are allowed to make linear extrapolations. . .'? Is this only valid for the last iteration step?*

The word manually has been removed to avoid confusion. It was used in an attempt to convey how the limits were imposed/hard-coded. The limits listed below necessarily apply to all retrieval steps to enable the forward model to function, because the forward model is used to evaluate the weighting functions. The limits, particularly in the case of water vapour, are generally extreme, and on such pixels where it is necessary (a handful out of a week's worth of orbits) there are typically other issues with the measurement that the forward model has a problem replicating. Such pixels are removed at point of quality control as they undoubtedly lead to either non-convergence or very large errors.

The text has been modified as follows:
". . .constrained in the FM by the physical limits that RTTOV will accept, or that are appropriate for the forward model. These are 0.01 to 800 DU for $SO_2$, 1e-6 to 16 times the column water amount predicted by ECMWF and a maximum cloud top height of 16 km (a conservative upper limit for tropopause height)."

*P6L25, 'The estimate accounts for inaccuracies that arise due to modelling the atmosphere at reduced spectral resolution, limited vertical resolution, inclusion of nonretrieved trace gases at a climatological level or their preclusion entirely, relative to a reference case.': What is meant with 'limited vertical resolution' and 'modelling .. at reduced spectral resolution'? How has this error been derived (line-by-line compared to band model)? How strong does this error depend on the atmospheric situation?*

Please also see response to P4L24 comment above.

Reduced spectral resolution refers to the fact that the RFM is a line by line model, but may be run at poorer resolution to represent something close to the way in which RTTOV represents spectral transmittances.

It is acknowledged that FME contributions may change depending upon the state, but even though the changes are expected to be small, characterising FME is a non-exhaustive process that can only estimate contributing sources of error. In this case, there are larger sources of error from elsewhere (such as incorrect height assignment, errors in representation of $SO_2$ profile in forward model or the presence of multi-layer optically thin cloud or ash) that are expected to be considerably more dominant.
Section 2.3 now states: "...limited vertical resolution (100 m versus 1 km as used in the forward model outside the region of the $SO_2$ perturbation),"

*P7L9, '100 DU': Could you put this number in perspective of the typical maximum column amounts e.g. after Pinatubo?*

The following has been added to the text:
"100 DU represents an $SO_2$ column from a large, explosive volcanic eruption. Pinatubo, for example, yielded column amounts of 350-500 DU (depending on instrument) after 24 hours which reduced to 100 DU after 7 days (Carn et al., 2005). The OMI instrument (see Table 1) captured column amounts of around 200 DU after the 2008 eruption of Kasatochi (Prata et al., 2010)."

*P7L16: How are the a-priori errors of the state vector element for water vapour set? Is this error only considered in the measurement space? Further, could you explain how the off-diagonal elements of the a-priori covariance matrix are set.*
The following text has been added to the relevant section: "The a priori error for water vapour is based on the variance of water vapour the ECMWF atmospheric training

profiles discussed above relative to the mean."

The error covariance of water with $SO_2$ is only considered in the measurement space, since it is applied to a channel that is sensitive to both water vapour and $SO_2$ it effectively absorbs the error covariance – it is mapped onto measurement space. As such there are no off-diagonal elements specified in the a priori covariance matrix. Provided the QC described is applied, they are not applicable.

*P8L1, Chapter 3.1, and Fig.1: Regarding the error bars shown in Fig. 1: Could you summarize which errors they contain? Have these errors been incorporated in the synthetic observations? (I assume no or only partially, otherwise the retrieval results should somehow scatter around these errors). Further, do the error bars represent 1 or 2-sigma values? The maximum value tested here seems to be 150 DU. Could you extend this range? You should also show, at which values the method fails and at which column amount of $SO_2$ the channel signal becomes saturated.*

This figure is used to demonstrate deficiencies in the Prata method and the linear behaviour of the OE column retrieval, as appropriate for the case study in particular.

The error bars show the retrieved error. The simulations were performed with simulated measurement noise (which is small) and FME. That is why an OE retrieval is so much more useful than a band model, such as the Prata model shown, which has no possibility of estimating error or quantifying uncertainty.

It is not the case that the maximum value tested is 150 DU, as can be seen from inspecting the far edge of the axis at 200 DU – a limit appropriate for the case study presented. The model fails above the training limit and in such a non-linear way that the authors feel it would be a distraction to show or dwell on this matter, as it is behaviour related to the complex inner mechanics of RTTOV which is itself a

comprehensive and complicated system. The channel becomes truly saturated above 1000 DU, far beyond the training limit, and as discussed elsewhere it would be the topic of future and non-trivial work to qualify the model behaviour for significantly larger eruptions.

This work is intended as a proof of concept, which the authors feel it demonstrates, rather than the definitive or comprehensive examination of the use of this instrument for all eruptions and for all HIRS/2 instruments. It is a worked demonstration applied to an eruptive event of significance, and as such the simulations and testing are all suitable for supporting both concept and case study.

The figure caption has been amended as follows:
"Retrievals based on simulations by a line-by-line model (RFM), with synthetic measurement noise. The error bars for the column retrieval are the retrieved errors."

*P8L25, 'Measurements were simulated for a plume at a range of altitudes from 8-18 km': However the caption in Fig. 2 says vice-versa: 'A 2 km thick triangular profile centred at 12 km is used to simulate measurements. The profile is then used in a retrieval with the retrieved height assigned to a range of altitudes.' Could you tell in which way the test retrievals have really been performed?*

This section (referring to what is now figure 3) now reads:
"Measurements were simulated for a plume at a range of altitudes from 8-18 km. Figure 3 shows the impact on the retrieved $SO_2$ column at a specified, fixed altitude of 12 km as a fraction of the true column at these altitudes. Errors range from typically $\pm$0-30
The caption for what is now figure 3 has been amended to read:
"A measurement was simulated for a volcanic plume of triangular profile centred at a range of altitudes, for a range of total column amounts. A retrieval is then performed

where the plume is assumed to be at 12 km. The fractional difference, or error, is plotted."

*P8L28: In this paragraph it is only referred to the Figure, however the results are not described. Please give also in the text at least some quantification of the resulting errors.*
Please see above.

*P9L1-3: It is not clear what is different here compared from the paragraph before.*

The text has been modified to make its purpose more clear. Specifically, it refers to a test of precision/accuracy of RTTOV vs RFM forward models, to show that it behaves in a similar way compared to the line-by-line model irrespective of $SO_2$ plume altitude: "The performance of the column fit was also directly assessed against a line-by-line model (RFM) for plume altitudes from 8 to 18 km (where the plume height assignment used in the retrieval was the same as that used in the measurement simulated by the RFM) and it was found that. . .".

*P9L6-13: Also here in the text some numbers (percentage error) should be mentioned. Further, could you explain, why there is such a large difference between the errors when the plume thickness is over- versus underestimated. Would this result not speak for application of a rather sharp profile in the retrieval to minimize the errors*

It is true that this suggests an underestimate of plume thickness would imply smaller errors than an overestimate. Further work would be required to establish an optimum thickness if there is one, particularly in relation to the vertical grid of the forward model, which may be a limiting factor. It is sufficient here to state that a profile that most resembles the true profile should be the best. In this case there is plenty

of ancillary information to give some indication of both plume thickness and plume altitude, namely lidar data. The actual error plume thickness cannot be known, but as with error in height assignment these simulations are a useful indicator to give confidence windows to whatever values might be retrieved.

Indicative numbers have been added to the text:
"The retrieval simulations suggest that errors are larger when the plume thickness is overestimated (typically 13
*P9L29: 'water vapour clouds' should perhaps read 'liquid water clouds' ?*
Text has been amended accordingly.

*P9L29, 'above 5 km': However in Fig. 4 the retrieval seems to be OK up to 8-9 km. Can you give an explanation why the retrieval has problems to fit cloud heights above a certain altitude. How much does it depend on the atmospheric situation (tropics vs mid/high latitudes)?*

The pertinent panel in figure 4 is the bottom right, which shows that deviations (errors) in retrieved water vapour column begin when a cloud is at 6 km. Poor fitting of water vapour leads to errors in the retrieval of $SO_2$, because the 7.3 micron channel is sensitive to both water vapour and $SO_2$. A cloud at 5 km shows no perceptible deviation, and it is likely that the threshold is somewhere in between in this case, which is the origin of the 5-6 km warning. The $H_2O$ weighting function of this channel peaks at 700hPa (but as the reviewer mentions this may vary slightly depending on the state, which is why a mid-latitude profile was used). The 6.8 micron channel weighting function peaks at 500 hPa. In the abstract 6 km was stated as this will definitely contribute an error to the retrieval of $SO_2$.
For clarity this has been corrected to 5 km since it is stated as "...above.." the given level.

*P10, chapter 3.4: What is the upper limit of the retrieved $SO_2$ (e.g. due to saturation effects)? This could also affect the total mass calculation in very dense plumes. Also some information about the convergence criteria of the retrieval and how many iterations are necessary are missing.*

This has not been tested with RTTOV. It would require the regression coefficients to be trained for much larger column amounts. Exploratory work with the RFM found that channel brightness temperature differences for a given change in $SO_2$ column amount become increasingly small over 600 DU for this channel with HIRS/2 on NOAA11. Since this is above column amounts observed even for Pinatubo, it has not been investigated exhaustively and is not referred to in the text. This may change slightly depending on the altitude of the $SO_2$, but would require further simulations and further work to investigate if this method were to be applied to a an eruption with very high $SO_2$ column amounts (Pinatubo or larger).

The text in section 2.1 has been modified as follows:

"The 7.3 $\mu$m channel is sensitive to both water vapour and $SO_2$. This channel may be said to saturate for $SO_2$ columns above 600 DU where significant increases in $SO_2$ result in small changes in channel BT below the envelope of the channel noise and other error terms."

P12L28: Could you state which $SO_2$ altitude profile has been used for the case study and how it has been derived. Is the resulting $SO_2$-altitude error included in the column errors in Fig. 5?

More work was done than has been stated to identify the plume altitude from Hudson, and some of it bears repeating to explain the origin of the $SO_2$ profile used in the model, before it is stated.

Lidar information has been used to identify the probable altitude of the plume of between 10-13km, peaking at around 12 km. This is considered to be a very reasonable estimate of the true shape of the $SO_2$ profile, given the considerable ancillary information available.

As stated, Figure 5 shows only the retrieval error. Given the amount of information about the plume height, it is unlikely that the plume height used in the retrieval is more than 1 km out. Figure 5 suggests that this would result in errors of not more than 10

The following has been added to the text:
"In addition, contemporary lidar measurements of the Hudson plume were made at the CSIRO (Commonwealth Scientificand Industrial Research Organisation) Division of Atmospheric Research, at Melbourne, Australia (38 S, 145 E) (Young et al., 1992, Barton et al. 1992). These measurements are sensitive to ash, sulphate aerosol and meteorological (water) cloud. The backscatter profiles tend to indicate peaks at around and above 20 km, and frequently at 10-13 km. The higher peak is attributed to aerosol from the Pinatubo eruption. Young et al. (1992) interpret the majority of observations that are thought to include Hudson material as the feature at 12 km in October, with variable cirrus at 10 km. It is reported by the authors that the plume was observed consistently from 28th August until December 1991 between 10 and 13 km, with a decreasing scattering ratio. The relative proportions that contribute to the backscatter measured are expected to be dominated by ash in the first few weeks after the eruption. Little ash is expected to be present after a month beyond the eruption, but by this time the vast majority of the $SO_2$ will have oxidised into aerosol. Whilst lidar is not sensitive to the presence of gaseous $SO_2$ inferences can be drawn from the height of the aerosol it eventually becomes. In this case the lidar information is considered to be a valuable starting point as a guide for estimating the cloud height of the $SO_2$, in the context of other information."

"Using all of this information, the Hudson plume is modelled as a triangular peaked profile with a baseline of 2 km between 11 and 13 km, peaking at 12 km."

*P14L5: To make this calculations more clear and give the reader a better feeling for the derived e-folding times and its possible uncertainties it would be necessary to plot derived daily masses after 17th August and show the fitted exponential decay line.*

This would be the case, but as mentioned in the text, due to the narrowness of the swath and the rate of motion of the plume, in addition to the presence ash in the first day after eruption, of total mass estimates on successive days do not follow a smooth curve. There are also several ways of estimating plume total mass, each have inherent issues associated with them that introduce error. Adding up total mass of the area represented by the footprint adds no information as other approaches like Kriging might, but in the case of HIRS/2 the calibration scanlines are missing, sometimes the plume is only partially sampled by a given orbit and there is movement between orbits. TOMS had the benefit of a wider swath, but HIRS/2 was able to sample both day and night so had more opportunity to monitor the plume. We found that gridding, which in theory might get around the problem of incomplete sampling resulted in total masses that were heavily dependent upon grid size and in most cases under-estimated the maximum plume mass compared to summing the areas represented by the satellite footprint. Just summing footprints results in a fairly noisy representation of the decay. It is a concern that to venture too far into this discussion is a distraction from the main point of the paper, which is to introduce the technique and demonstrate its effectiveness using a case study. We feel that adding a figure would require significantly more discussion about this issue than is warranted here. Furthermore, while the majority of the $SO_2$ was released by Hudson on 12th August, about 30

We have now mentioned some of these further points in this section:
"...be overly-generous bounds by this method. This case is complicated by the fact

that about 30
The following text has been added:
"In reality the total mass observed does not decay smoothly, but has noise due to the fact that the plume is not always perfectly sampled, and the number of retrieved pixels excluded due to the presence of high or thick cloud or ash varies."

Also:
"More recently, Carn et al. (2016) estimated the e-folding time of Cerro Hudson to be 7 days, based on mass estimates from TOMS (Constantine et al., 2000). They attribute this anomalously short e-folding time to the late southern hemisphere winter timing of the eruption. However, since Constantine et al., (2000) estimate nearly twice the total mass (4000kT) than that observed by HIRS/2 in this work (and the subsequent TOMS algorithm discussed here) it is possible that the inconsistency in e-folding times could be due to an over-estimate of initial erupted mass from the original TOMS algorithms. Total mass estimates (and therefore e-folding time estimate) would be improved greatly in accuracy if the HIRS/2 instruments aboard NOAA10 and NOAA12 that were also present were used to result in very comprehensive sampling of this eruption."

*P14K19, '2300 +- 600 kT': How has the error of 600 kT been calculated?*

This is the average retrieval error to appropriate significant figures. The text has been amended as follows:
"This OE column retrieval finds a new total erupted mass estimate for the 1991 eruption of Cerro Hudson of 2300 ± 600 kT from the HIRS/2 instrument aboard NOAA11, where the error is the retrieved error."
Table 3 now also explicitly mentions this in its caption (see below).

*P14L22: Please give also the total masses (including errors) of TOMS, Carn et al., 2016 and Prata et al., 2003.*

The text has been modified to highlight the mass and origin of mass calculated, with further discussion added regarding the inconsistency in e-folding time that is thought to result from the Carn comaparison in particular. No errors are given in that text, but as their origin is the Constantine paper, it is mentioned elsewhere in the text that they estimated their retrieval error to be of the order of 30

The Prata et al., (2003) method estimates an error of 5

The following text has been added to the introduction of the Prata method: "Indeed, it is not possible to formally quantify error of mass estimates from this method as it currently stands."

The caption to what is now Figure 2 has had the following added: "No error estimates are possible for the Prata fit method."

Table 3 (showing comparative mass estimates) has had two comments regarding errors added:

"1Constantine et al. (2000), with errors estimated to be circa 30 %."

"4This work, with retrieved error."

*Technical comments:*
*P1L31, 'The TOVS instrument': But in the sentence before it is explained as a suite of instruments.*

This has been clarified in the text.

*P1L32, 'TIROS': Is written 'TIrOS' in L28.*

These have been unified.

*P4L21, 'Table 1': Shouldn't this read 'Table 2'?*

This has been corrected.

*P11L7: '(Constantine et al. 2000)' to '(Constantine et al., 2000)'*

This has been corrected.

*P12L14: 'verses' to 'versus'*
This has been corrected.

*P13L4: '317 channel' to '317 nm channel'*
Corrected.

*P13L5: '340 to' to 'channel at 340 nm to'*
Corrected.

*P14L3: Should the formula not read: N(t) = N0 exp (- lambda t)*
Corrected.

*P15L15: 'satellites' to 'satellite'*
Corrected.

*P23Fig3: '11.5, 12, 13 km' should read '11.5, 12, 12.5 km'*
Corrected.